# ROCK and the actomyosin network control biomineral growth and morphology during sea urchin skeletogenesis

Eman Hijaze[1], Tsvia Gildor[1], Ronald Seidel[2†], Majed Layous[1], Mark Winter[3], Luca Bertinetti[2], Yael Politi[2], Smadar Ben-Tabou de-Leon[1]*

[1]Department of Marine Biology, Leon H. Charney School of Marine Sciences, University of Haifa, Haifa, Israel; [2]B CUBE Center for Molecular Bioengineering, Technische Universität Dresden, Dresden, Germany; [3]Department of Electrical Engineering, Computer Science and Mathematics, Technische Universiteit Delft, Delft, Netherlands

*For correspondence:
sben-tab@univ.haifa.ac.il

Present address: †Section Biomedical Imaging, Molecular Imaging North Competence Center (MOIN CC), Dept. Radiology and Neuroradiology, University Medical Center Kiel, Kiel University, Kiel, Germany

Competing interest: The authors declare that no competing interests exist.

**Abstract** Biomineralization had apparently evolved independently in different phyla, using distinct minerals, organic scaffolds, and gene regulatory networks (GRNs). However, diverse eukaryotes from unicellular organisms, through echinoderms to vertebrates, use the actomyosin network during biomineralization. Specifically, the actomyosin remodeling protein, Rho-associated coiled-coil kinase (ROCK) regulates cell differentiation and gene expression in vertebrates' biomineralizing cells, yet, little is known on ROCK's role in invertebrates' biomineralization. Here, we reveal that ROCK controls the formation, growth, and morphology of the calcite spicules in the sea urchin larva. ROCK expression is elevated in the sea urchin skeletogenic cells downstream of the Vascular Endothelial Growth Factor (VEGF) signaling. ROCK inhibition leads to skeletal loss and disrupts skeletogenic gene expression. ROCK inhibition after spicule formation reduces the spicule elongation rate and induces ectopic spicule branching. Similar skeletogenic phenotypes are observed when ROCK is inhibited in a skeletogenic cell culture, indicating that these phenotypes are due to ROCK activity specifically in the skeletogenic cells. Reduced skeletal growth and enhanced branching are also observed under direct perturbations of the actomyosin network. We propose that ROCK and the actomyosin machinery were employed independently, downstream of distinct GRNs, to regulate biomineral growth and morphology in Eukaryotes.

## eLife assessment

This **valuable** study addresses the role of Rho-associated coiled-coil kinase (ROCK) and the cytoskeleton in the initiation and growth of the calcified endoskeleton of sea urchin embryos. Perturbation by two independent approaches (a morpholino and a selective inhibitor) provides **convincing** evidence that ROCK participates both in actomyosin regulation and in the gene regulatory network that controls skeletogenesis. Exciting areas of future work will be to elucidate the mechanisms by which ROCK influences gene expression and to further dissect the role of the cytoskeleton in mineralization.

## Introduction

Biomineralization is the process in which organisms from the five kingdoms of life, use minerals to produce shells, skeletons, and teeth that protect and support them (*Weiner and Addadi, 2011*; *Murdock and Donoghue, 2011*; *Knoll, 2003*; *Gilbert et al., 2022*). Recent studies suggest that

biomineralization evolved by the phylum-specific co-option of ancestral GRNs that drove the construction of distinct organic scaffolds (*Gilbert et al., 2022*; *Ben-Tabou de-Leon, 2022*; *Morgulis et al., 2019*; *Murdock, 2020*; *Hojo et al., 2016*; *Fraser et al., 2009*) and by the evolution of specialized sets of biomineralization proteins (*Knoll, 2003*; *Murdock, 2020*; *Evans, 2019*; *Khor and Ettensohn, 2017*; *Kawasaki et al., 2009*; *Drake et al., 2013*). This explanation is in line with the dissimilar GRNs and biomineralization proteins that drive biomineralization in different phyla (*Ben-Tabou de-Leon, 2022*; *Kawasaki et al., 2009*; *Tucker and Sharpe, 2004*). Yet, there are common cellular processes required for mineralization in highly diverse phyla (*Gilbert et al., 2022*), suggesting that distinct upstream GRNs might have recruited common cellular and molecular mechanisms to drive biomineralization.

The actomyosin network was shown to play a role in biomineralization in various Eukaryote models, from unicellular organisms to vertebrates' bones and teeth (*Langer et al., 2010*; *Durak et al., 2017*; *Tyszka et al., 2019*; *Tesson and Hildebrand, 2010a*; *Tesson and Hildebrand, 2010b*; *Strzelecka-Kiliszek et al., 2017*; *Qin et al., 2020*; *Huang et al., 2018*). There is a tight association between actin filaments and the biomineralization compartment in diatoms (*Tesson and Hildebrand, 2010b*) and a highly dynamic actin organization that forms the biomineralization compartment in foraminifera (*Tyszka et al., 2019*). Perturbations of actin polymerization result in severely deformed shells and inhibition of shell shedding in coccolithophores (*Langer et al., 2010*; *Durak et al., 2017*) and diatoms (*Tesson and Hildebrand, 2010a*). In vertebrates, Rho GTPases and ROCK, regulate chondrocytes, osteoblasts, and odontoblasts differentiation and affect gene expression in these biomineralizing cells (*Strzelecka-Kiliszek et al., 2017*; *Qin et al., 2020*; *Huang et al., 2018*). However, the roles of actomyosin remodeling in controlling mineral deposition and shaping biomineral morphology, are still unclear.

Sea urchin larval skeletogenesis provides an attractive model to study the role of the actomyosin network in biomineralization. Sea urchin larval skeletons are made of two frameworks of interconnected calcite rods, termed 'spicules' that are generated by the skeletogenic cells (*Morgulis et al., 2019*; *Oliveri et al., 2008*). To make the spicules, the skeletogenic cells form a ring with two lateral skeletogenic clusters and fuse through their filopodia forming a pseudopodia cable that links them into a syncytium (*Ettensohn and Dey, 2017*; *Gildor et al., 2021*). The mineral is concentrated in the form of amorphous calcium carbonate (ACC) inside intracellular vesicles (*Kahil et al., 2020*; *Kahil et al., 2023*). The vesicles are secreted into the biomineralization compartment generated in the skeletogenic cell clusters forming two triradiate spicules (Figure 8A, *Kahil et al., 2020*; *Vidavsky et al., 2014*). The tubular biomineralization compartment, also called the spicule cavity, elongates within the pseudopodia cable by localized mineral deposition at the tip of each rod (*Wilt et al., 2008*).

The GRN that controls sea urchin skeletogenesis is known in great detail and is highly similar to the GRN that controls vertebrates' vascularization, suggesting a common evolutionary origin of these two tubulogenesis programs (*Morgulis et al., 2019*; *Oliveri et al., 2008*; *Adomako-Ankomah and Ettensohn, 2013*). As the spicules elongate, the expression of key regulatory and biomineralization-related genes becomes restricted to the skeletogenic cells proximal to the growing tips, possibly to promote mineral deposition at these sites (*Sun and Ettensohn, 2014*; *Tarsis et al., 2022*; *Morgulis et al., 2021*). Localized gene expression is regulated by signaling cues such as the VEGF signaling (*Adomako-Ankomah and Ettensohn, 2013*; *Sun and Ettensohn, 2014*; *Tarsis et al., 2022*; *Duloquin et al., 2007*). However, how the skeletogenic GRN drives spicule formation and localized mineral deposition at the tips of the rods is poorly understood.

Previous works suggest that the actomyosin network is essential for sea urchin skeletal growth. Calcium-bearing vesicles perform an active diffusion motion in the skeletogenic cells with a diffusion length that inversely correlates with the strength and activity of the actomyosin network (*Winter et al., 2021*). Actin filaments are formed around the spicule (*Winter et al., 2021*; *Stepicheva et al., 2017*) and F-actin signal is enriched at the tips of the growing skeletal rods in skeletogenic cell culture (*Moreno et al., 2019*). Genetic and pharmacological perturbations of the GTPase, CDC42, prevent the formation of filopodia in the skeletogenic cells and inhibit spicule formation and elongation (*Sepúlveda-Ramírez et al., 2018*). Pharmacological perturbations of ROCK prevent spicule formation (*Croce et al., 2006*) and genetic perturbations of Rhogap24l/2 result in ectopic spicule splitting (*Morgulis et al., 2019*). Despite these characterizations, little is known about the role of the actomyosin machinery in regulating sea urchin biomineral growth and morphology.

Here, we study the role of ROCK and the actomyosin network in the sea urchin *Paracentrotus lividus (P. lividus)*. Our findings reveal the critical role of ROCK and the actomyosin network in multiple aspects of sea urchin biomineralization, suggesting a common use of these factors in Eukaryote biomineralization, downstream of distinct GRNs.

## Results

### ROCK is enriched in the skeletogenic cells depending on VEGF signaling

We sought to study the spatial expression of the ROCK protein and its regulation by VEGF signaling, a prominent regulator of sea urchin skeletogenesis (*Morgulis et al., 2019*; *Adomako-Ankomah and Ettensohn, 2013*). The sequence of ROCK and especially, its functional domains are highly conserved between human and sea urchin (*Figure 1—figure supplement 1*, *Marlétaz et al., 2023*; *Arshinoff et al., 2022*; *Telmer et al., 2024*). According to RNA-seq data measured in *P. lividus*, ROCK is a maternal gene that degrades after the maternal to zygotic transition and picks up again after hatching >14 hr post fertilization (hpf), (*Figure 1—figure supplement 2A*). We used a commercial antibody that recognizes human ROCK protein to test ROCK expression in *P. lividus* embryos. We first used western blot to detect ROCK expression in control and under VEGFR inhibition at the time of spicule formation and during skeletal elongation (Axitinib treatment, ~22 hpf, 27hpf, and 33hpf, *Figure 1—figure supplement 2D*). The antibody detected a ~150 kDa protein (*Figure 1—figure supplement 2D*) that is the predicted size for *P. lividus* ROCK protein (153 kDa). VEGFR inhibition marginally increased the overall level of ROCK at 22hpf, but did not affect it at 27hpf and 33hpf (*Figure 1—figure supplement 2E*). Yet, this measurement was done on proteins extracted from whole embryos, of which the skeletogenic cells, where VEGFR is active, are less than 5% of the total cell mass (*Massri et al., 2021*). We, therefore, wanted to study the spatial expression of ROCK and specifically, its regulation in the skeletogenic cells.

We studied the spatial distribution of ROCK protein using ROCK antibody and the skeletal cell marker, 6a9, that marks the skeletogenic syncytium membrane, in control and under VEGFR inhibition (*Ettensohn and McClay, 1988*, *Figure 1*). We quantified ROCK signal in the skeletogenic cells compared to the neighboring ectodermal cells in both conditions, at the three timepoints mentioned above. In the three timepoints, under VEGFR inhibition, the skeletogenic cells do not migrate to their proper positions in the anterolateral and post-oral chains, as previously reported (*Morgulis et al., 2019*; *Adomako-Ankomah and Ettensohn, 2013*; *Duloquin et al., 2007*). At 22hpf, ROCK expression in the skeletogenic cells is mildly enriched compared to the neighboring ectodermal cells, and this enrichment is not observed under VEGFR inhibition (*Figure 1A–F*). At 27hpf, ROCK enrichment in the skeletogenic cells increases, but similar enrichment is observed under VEGFR inhibition (*Figure 1G–L*). At 33hpf, ROCK enrichment in the skeletogenic cells is most apparent (*Figure 1M–O*) and depends on VEGF signaling (*Figure 1P-S*). At both 27hpf and 33hpf in control embryos, ROCK seems localized near the skeletogenic cell membranes, which further supports its activation in these cells, since ROCK activation leads to its localization to the cell membranes in other systems (*Chen et al., 2014*). Overall, this data demonstrates that ROCK expression is elevated in the skeletogenic cells and this enrichment strengthens with skeletal elongation and depends on VEGF signaling.

### ROCK activity in the skeletogenic cells controls spicule initiation, growth, and morphology

Next, we studied the role of ROCK in sea urchin skeletogenesis using genetic perturbations. We downregulated the expression of ROCK by the injection of two different translation morpholino antisense oligonucleotides (MASOs). ROCK MASO-1 matches the start of the translation region, ROCK MASO-2 matches the 5' region upstream the start of translation, and random MASO was used as a control. Embryos injected with either ROCK MASO-1 or MASO-2 show reduced ROCK signal at 33hpf, supporting the downregulation of the ROCK protein using these MASO's (*Figure 2—figure supplement 1B, F, J, N, R, V*).

Embryos injected with ROCK MASO-1 or MASO-2 show strongly reduced skeletons at two days post fertilization (2dpf, *Figure 2C and D*) and embryos injected with ROCK MASO-2 show an additional phenotype of the severely branched skeleton at this time (*Figure 2B*). The additional branching

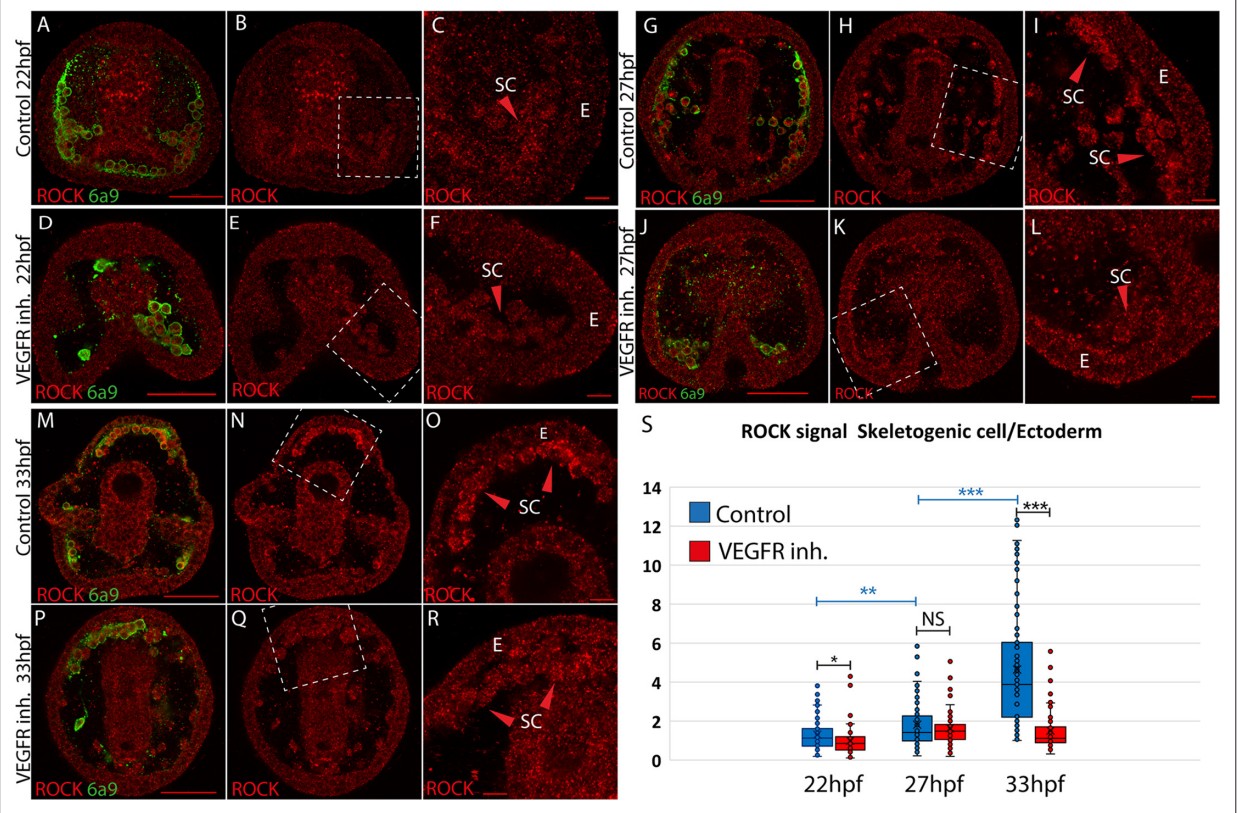

**Figure 1.** Rho-associated coiled-coil kinase (ROCK) enrichment in the skeletogenic cells increases with time and depends on vascular endothelial growth factor (VEGF) signaling. (**A–R**) ROCK immunostaining at different timepoints in control and under VEGFR inhibition (150 nM Axitinib). In each condition and time point, the left image shows ROCK immunostaining and the skeletogenic cell marker, 6a9 (**A, D, G, J, M**, and **P**). The middle image shows ROCK immunostaining alone in the whole embryo (**B, E, H, K, N**, and **Q**). The right image shows the enlargement of the white rectangle region of the middle image (**C, F, I, L, O, R**). Scale bar in whole embryo is 50 µm and in enlargements is 10 µm. E – ectoderm, SC – skeletogenic cells. (**S**) Quantification of the ratio between ROCK signal/area in the skeletogenic cells compared to the ectodermal cells (see methods for details). Each box plot shows the average marked in x, the median, the first and the third quartiles (edges of boxes), and all experimental measurements (dots). Experiments were performed in three independent biological replicates and in each condition, at least 33 embryos were measured. Statistical significance was measured using paired two-tailed t-test where, * indicates p<0.05, ** indicates p<0.005, and *** indicates p<0.0005.

The online version of this article includes the following source data and figure supplement(s) for figure 1:

**Source data 1.** Measurements of Rho-associated coiled-coil kinase (ROCK) signal in the skeletogenic cell vs. the ectoderm, in control and VEGFR inhibition.

**Figure supplement 1.** Sea urchin Rho-associated coiled-coil kinase (ROCK) sequence and conserved domains.

**Figure supplement 2.** Time course of Pl-ROCK, immunostaining, and western blot of Rho-associated coiled-coil kinase (ROCK) antibody in sea urchin embryos.

**Figure supplement 2—source data 1.** RNA-seq measurement of Pl-ROCK temporal expression.

**Figure supplement 2—source data 2.** Quantification of the western blot of Rho-associated coiled-coil kinase (ROCK) antibody in control and VEGFR inhibition.

phenotype and the larger percentage of affected embryos in ROCK MASO-2 are probably due to its higher efficiency resulting from its more 5' location relative to the translation start codon (*Figure 2E*). Thus, the genetic perturbations of ROCK expression indicate that sea urchin ROCK is important for skeletal elongation and normal branching patterns.

To elucidate the dependence of ROCK phenotypes on ROCK activity level and identify its effect on different stages of skeletogenesis, we tested the skeletogenic phenotypes of different concentrations of ROCK inhibitor, Y27632, applied at different timepoints (*Figure 2—figure supplement 2A*). Y27632 binds to the ATP site of ROCK's kinase domain and prevents its activity (*Jacobs et al., 2006*). Y27632 affinity to ROCK is 100 times higher than its affinity to other kinases, such as PKA and PKC (*Narumiya et al., 2000*). The amino-acids to which Y27632 binds, are conserved in the sea urchin

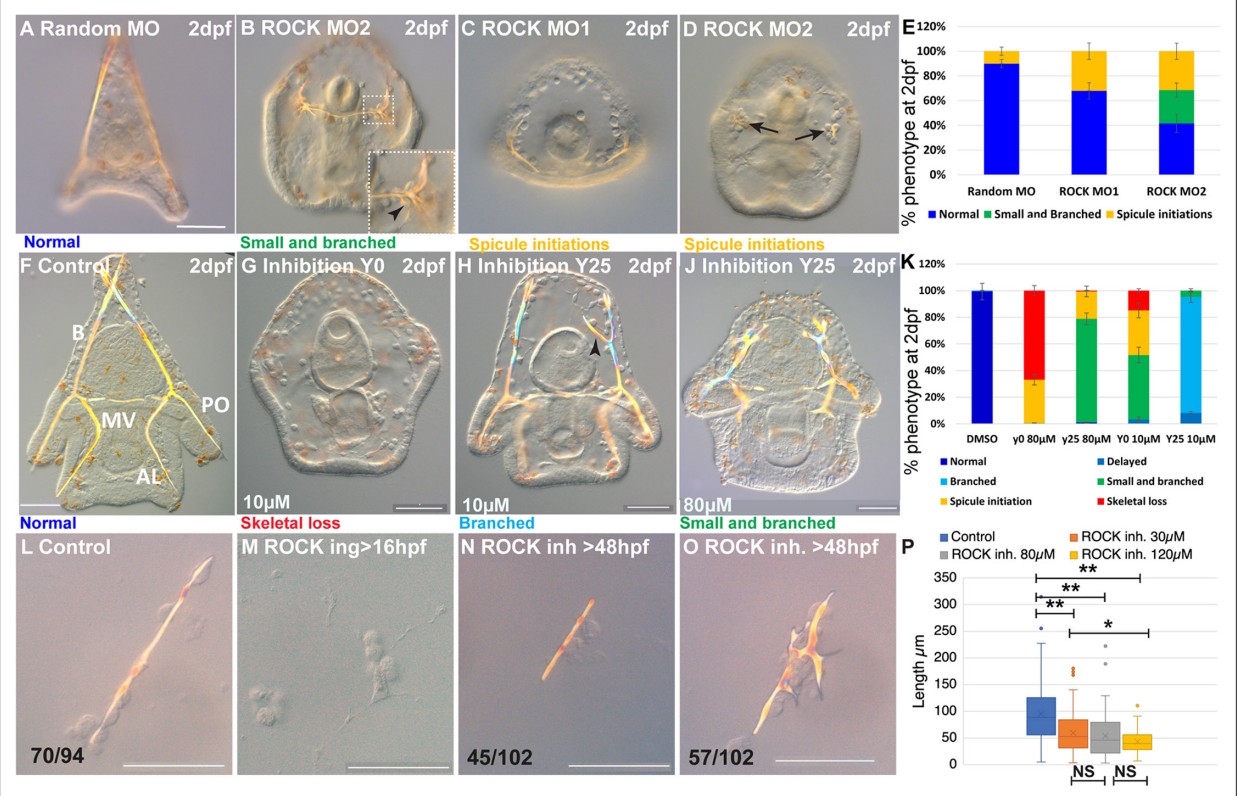

**Figure 2.** Rho-associated coiled-coil kinase (ROCK) activity is essential for spicule formation, normal elongation, and branching in whole embryos and in skeletogenic cultures. (A–E) genetic perturbation of ROCK translation using two different morpholino anti-sense oligonucleotides (MASOs) observed at 2dpf. (A) Control embryo injected with Random MASO. (B) Embryo injected with ROCK MO-2 shows ectopic spicule branching. (C, D) Embryos injected with ROCK MO-1 or MO-2 show spicule initiations. (E) Summary of MASO injection phenotypes based on 4–6 independent biological replicates. (F–K) Pharmacological perturbations of ROCK activity using 10 μM and 80 μM of the inhibitor Y27632 were observed at 2dpf. (F) Representative control embryo with normal skeletal rods, B, body; AL, anterolateral; PO, post-oral and MV, midventral. (G) Complete skeletal loss in embryos treated continuously with 10 μM ROCK inhibitor. (J) Reduced skeletal growth and enhanced ectopic branching in embryos where 10 μM ROCK inhibitor was added at 25hpf. (J) Small spicules with enhanced ectopic branching in embryos where 80 μM ROCK inhibitor was added at 25hpf. (K) Summary of perturbation phenotypes based on three to eight biological replicates for each treatment. See additional concentrations, phenotypes, and summary in *Figure 2—figure supplement 2* and *Supplementary file 1a*. (L–O) Representative spicules from skeletogenic cell cultures in control and under 30 μM Y27632 at 72hpf. (L) linear spicule in control culture, (M) Y27632 addition at 16hpf, before spicule initiation, completely blocks spiculogenesis. (N, O) Y27632 addition after spicule initiation, at 48hpf, reduces spicule elongation (N), and enhances branching (O). (P) Quantification of spicule length in control and ROCK inhibition (>48 hpf) at 72hpf. *p<0.05, **p<0.001, Kruskal-Wallis non-parametric test. Results are based on three biological repeats for each treatment, except from 120 μm that was done in two biological repeats. Scale bars are 50 μm. In L, N, and O, the numbers at the bottom indicate the number of spicules that show this phenotypes (left) over all observed spicules (right).

The online version of this article includes the following source data and figure supplement(s) for figure 2:

**Source data 1.** Phenotypes of Rho-associated coiled-coil kinase (ROCK) morpholino anti-sense oligonucleotides (MASO) experiments presented in *Figure 2E*.

**Source data 2.** Phenotypes of Rho-associated coiled-coil kinase (ROCK) inhibitor experiments presented in *Figure 2K* and *Figure 2—figure supplement 2Y*.

**Source data 3.** Measurements of spicule length in skeletogenic cell cultures in control and Rho-associated coiled-coil kinase (ROCK) inhibition, presented in *Figure 1P*.

**Figure supplement 1.** Rho-associated coiled-coil kinase (ROCK) and F-actin expression in control and ROCK morpholino anti-sense oligonucleotides (MASO) injections at 33hpf.

**Figure supplement 2.** Rho-associated coiled-coil kinase (ROCK) inhibition phenotypes for different treatments and concentrations.

**Figure supplement 3.** Time course of the effect of Rho-associated coiled-coil kinase (ROCK) inhibition after 25hpf.

ROCK protein, supporting its specific inhibition of this protein (*Figure 1—figure supplement 1C*). Y27632 had been used in cell cultures and live embryos from vertebrates to *Drosophila,* in concentrations between 10–100 μM (*Narumiya et al., 2000*; *Segal et al., 2018*; *Rousso et al., 2016*). In the sea urchin embryo, a concertation of 75 μM of Y27632 was reported to completely block skeleton formation (*Croce et al., 2006*) and a concentration of 100 μM was shown to delay gut invagination (*Beane et al., 2006*). Therefore, we tested the range of 10–80 μM Y27632 applied before or after spicule initiation (*Figure 2—figure supplement 2A*).

Continuous ROCK inhibition beginning at egg fertilization, resulted in significant skeletogenic phenotypes that are dose-dependent (*Figure 2*, *Figure 2—figure supplement 2*). Continuous ROCK inhibition using 80 μM Y27632, did not affect skeletogenic cell migration but eliminated skeletal formation at 27hpf (*Figure 2—figure supplement 2B, C*). At 2dpf, complete skeletal loss is detected under continuous ROCK inhibition in all concentrations, ranging from 15% of the embryos exposed to 10 μM to 67% of the embryos exposed to 80 μM Y27632 (*Figure 2G and K*, in agreement with *Croce et al., 2006*). The rest of the embryos exposed to continuous ROCK inhibition show either spicule initiations or small spicules with enhanced ectopic branching, with the percentage depending on Y27632 concertation (*Figure 2K*, *Figure 2—figure supplement 2K*). Except from the skeletogenic phenotypes, the overall embryonic development of embryos exposed to Y27632 in all concentrations seems normal (*Figure 2G and J*). Similar results are observed when adding the inhibitor at 20hpf, with the distribution of phenotypes that depends on the concentration (*Figure 2—figure supplement 2I, K*). Importantly, skeletal loss, spicule initiation, and ectopic branching were not observed under PKC or PKA inhibition, that resulted in much milder skeletogenic phenotypes (PKC) or no skeletogenic phenotype (PKA, *Mitsunaga et al., 1990a*; *Mitsunaga et al., 1990b*), supporting the selective inhibition of ROCK by Y27632. Altogether, continuous ROCK inhibition results with severe skeletogenic phenotype ranging from complete skeletal loss to small spicules with ectopic branching, with ratios that depend on the inhibitor concentration.

The addition of the inhibitor at 25hpf, after spicule initiation, results in a majority of embryos showing ectopic spicule branching (*Figure 2H–K*, *Figure 2—figure supplement 2F-H, K*). The reduction of skeletal growth rate and ectopic branching can be observed a few hours after the addition of ROCK inhibitor (*Figure 2—figure supplement 3*, 30 μM). Washing the inhibitor at the highest concertation after 25hpf results in partial recovery of skeletogenesis, with normal or mildly delayed skeletons, indicating that ROCK inhibition is reversible (*Figure 2—figure supplement 2E, K*).

Overall, the genetic and pharmacological perturbations of ROCK result in a sever reduction of skeletal growth and enhanced skeletal branching, but only continuous ROCK inhibition leads to complete skeletal loss (*Figure 2G and K*). Immunostaining of fertilized eggs clearly shows that ROCK protein is maternal, in agreement with our RNA-seq data (*Figure 1—figure supplement 2A–C*). The injected MASO's cannot interfere with the maternal ROCK protein whereas the inhibitor affects the activity of both maternal and zygotic ROCK, which could underlie the absence of skeletal loss in the genetic perturbations. Indeed, the ROCK signal is only moderately reduced in ROCK MASO-injected embryos at 33hpf (*Figure 2—figure supplement 1F, J, N*), demonstrating the partial penetration of the MASO. Thus, our genetic and pharmacological perturbations reconcile, and indicate that ROCK activity is necessary for spicule formation, skeletal elongation, and normal branching pattern.

To test if ROCK skeletogenic phenotypes are due to its activity specifically in the skeletogenic cells, we inhibited ROCK in a culture of isolated skeletogenic cells (*Figure 2L–P*). The addition of ROCK inhibitor to the skeletogenic cell culture at 16hpf, before the spicules form, completely abolished spicule formation (*Figure 2M*). The addition of the inhibitor at 48hpf, after spicule formation, resulted in significantly shorter spicules (*Figure 2L and N*) and increased branching compared to control spicules (*Figure 2O*). The reduction of the spicule length becomes more significant with increasing concentration of the ROCK inhibitor (*Figure 2P*). Notably, skeletal loss and ectopic spicule branching were not observed in PKA or PKC inhibition in skeletogenic cell cultures (*Mitsunaga et al., 1990b*), further supporting the selective inhibition of ROCK by Y27632. ROCK skeletogenic phenotypes in the isolated skeletogenic cell culture are similar to its phenotypes in whole embryos, verifying that ROCK activity in the skeletogenic cells is essential for spicule formation, normal elongation, and prevention of branching.

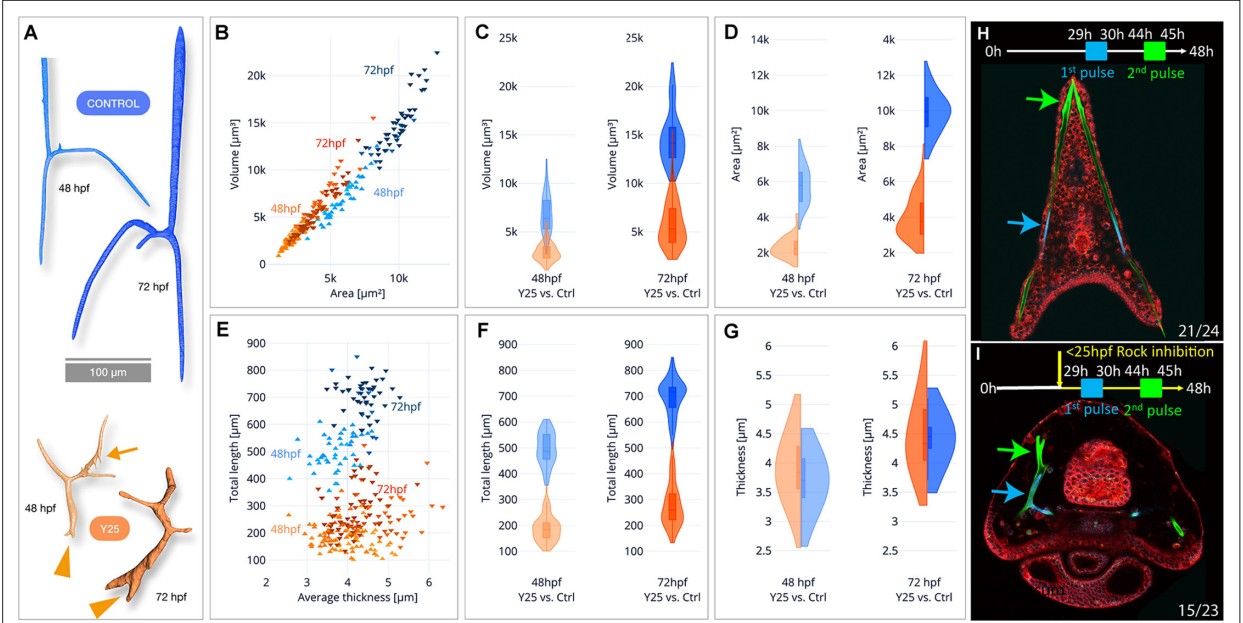

**Figure 3.** Synchrotron radiation micro-computed tomography (SR-μCT) studies of Rho-associated coiled-coil kinase (ROCK)-inhibited spicules show a reduction in skeletal volume, surface area, and total length, but not thickness and two-pulse calcein shows loss of tip-dominance. (**A**) Exemplary 3D-renderings of control spicules (top, blue) and spicules where 40 μM of ROCK inhibitor was added at 25hpf (bottom, orange), dissected at 48hpf and 72hpf. Arrowheads point to tip splitting and arrows at 48hpf point to spicule dripping at the back. (**B**) Spicule volume vs. area for control and ROCK-inhibited spicules at 48 and 72 hpf. Each data point represents a single spicule. (**C–D**) Frequency distributions for volume and surface area of control and ROCK-inhibited spicules at 48 hpf and 72 hpf (left and right violin plots, respectively). (**E**) Spicules' total branch length and average thickness for control and ROCK-inhibited spicules at 48 and 72 hpf. (**F–G**) Frequency distributions for spicule lengths and thickness of control and ROCK-inhibited spicules were dissected at 48hpf and 72hpf. (**H, I**) Calcein two-pulses experiment. Embryos were exposed to blue calcein at 29-30hpf, to green calcein at 44-45hpf, and stained with FM4-64 membrane marker (red) prior to image acquisition at 48hpf. (**H**) Control embryo, (**G**) Embryo where 30 μM of Y27632 was added at 25hpf. The experiments were done in three biological replicates and the numbers at the bottom indicate the number of embryos that show this phenotype out of all embryos scored.

The online version of this article includes the following source data and figure supplement(s) for figure 3:

**Source data 1.** Measurements of spicule length, width, surface area, and volume in μ-CT.

**Figure supplement 1.** Sample preparation for synchrotron radiation micro-computed tomography (SR-μCT) data acquisition and analysis.

**Figure supplement 2.** Effect of Rho-associated coiled-coil kinase (ROCK) inhibition on calcein stained area and vesicle number.

**Figure supplement 2—source data 1.** Quantification of vesicle number and area in control and Rho-associated coiled-coil kinase (ROCK) inhibition.

## ROCK inhibition reduces spicule volume, surface area, and total length, but not thickness (SR-μCT)

To quantify the effect of ROCK inhibition on mineral deposition and skeletal shape we used synchrotron radiation micro-computed tomography (SR-μCT) that enables three-dimensional visualization and quantification of spicule morphology at 2 and 3dpf (*Figure 3—figure supplement 1*, 40 μM Y27632, see methods for details). The aberrant morphology of ROCK-inhibited spicules is visible at both developmental time points (*Figure 3A*). The ectopic branching occurs both at the tips where tip-splitting is detected (arrowheads, *Figure 3A*), as well as in the back, where small spicule extensions, or mineral 'dripping' are observed (arrow at 48hpf, *Figure 3A*). This demonstrates that the regulation of mineral deposition is perturbed both at the tip and in the back of the spicules under ROCK inhibition.

When compared to control spicules, ROCK inhibition leads to a ~2.5 fold reduction of the spicule volume and surface area in both 2 and 3dpf (*Figure 3B–D*, see *Supplementary file 1b* for measured values and standard deviations and *Figure 2—figure supplement 1* for statistical analyses). The average rate of spicule growth between 48hpf and 72hpf in control embryos is 325.5 μm³/hr, and reduces to 119.8 μm³/hr under ROCK inhibition. Hence, the rate of mineral deposition is significantly reduced under ROCK inhibition.

To understand the specific effect of ROCK inhibition on spicule elongation vs. thickening, we compared the total length and the mean thickness of the spicules between control and ROCK inhibition (*Figure 3E–G*). The total length of control spicules was on average about 2.5 times longer than the length under ROCK inhibition in both 2dpf and 3dpf (*Figure 3E and F*, *Supplementary file 1b and c*). In contrast to its effect on the spicule length, ROCK inhibition caused a minor increase of the spicule thickness at 2dpf, and at 3dpf the thickness was not affected by the inhibition (*Figure 3E–G*, *Supplementary file 1b and c*). These SR-μCT measurements indicate that ROCK activity plays a crucial role in controlling the rate of mineral deposition at the tips of the rods and spicule elongation, and is not affecting spicule thickening.

## ROCK activity does not control mineral uptake but is required for 'tip-dominance'

To test the effect of ROCK inhibition on mineral intake, we used the fluorescent chromophore, calcein that binds to divalent ions including $Ca^{+2}$, and is widely used to track the calcium pathway in biomineralization (*Morgulis et al., 2019*; *Vidavsky et al., 2014*; *Morgulis et al., 2021*; *Winter et al., 2021*). We measured the number of calcein-stained vesicles per area as well as the number of pixels of calcein signal per area in the skeletogenic cells, in control and under ROCK inhibition at 1dpf and 2dpf (*Figure 3—figure supplement 2A–E*). Continuous ROCK inhibition does not change the number of calcium-bearing vesicles at 1dpf, but the number of calcein-stained pixels significantly increases in this condition, possibly indicating that the vesicle volume is larger (*Figure 3—figure supplement 2F, G*). At 2dpf, the number of calcein-stained pixels is similar in control and in continuous ROCK inhibition, suggesting that the overall calcium content does not change in this condition (*Figure 3—figure supplement 2H, I*). The number of calcein-stained vesicles is, however, decreased under ROCK inhibition, possibly indicating that there are fewer vesicles with larger volume. Addition of ROCK inhibitor at 25hpf affects neither the number of calcein-stained vesicles nor the number of calcein-stained pixels in the skeletogenic cells. Together, these measurements show that ROCK is not required for the uptake of calcium into the cells. Therefore, the skeletal phenotypes occurring under ROCK inhibition are not related to a decrease in calcium uptake or shortage in intracellular calcium, but are due to defects in mineral processing within the cells and in the mineral deposition.

To monitor the effect of ROCK inhibition on mineral deposition rate and distribution, we applied two pulses of different calcein dyes (*Descoteaux et al., 2023*): blue calcein was added at 29hpf and washed after 1 hr, followed by green calcein added at 44hpf and washed after 1 hr. Hence, blue calcein marks the initial triradiate region and green calcein marks the edges of the spicules that were recently deposited blue and green arrows in *Figure 3H I*. The dye FM4-64 was used to mark the cell membranes. Under ROCK inhibition, the green labeled region is much closer to the blue labeled region compared to the control, due to the reduction in skeletal elongation rate, in agreement with our SR-μCT measurements. However, while in control embryos each body rod has a single tip stained in green calcein, under ROCK inhibition one of the body rods has two tips and both are stained in green. This indicates that mineral is being deposited in both edges, and the mechanism that prevents the growth of multiple tips in each rod, enabling 'tip-dominance,' (in analogy to plant stem apical dominance), is disrupted under ROCK inhibition.

## The activity of the actomyosin network is essential for normal spicule elongation and inhibition of ectopic branching

In other systems, the major roles of ROCK are to control F-actin polymerization and MyoII activation (*Liu et al., 2018*; *Julian and Olson, 2014*; *Lecuit et al., 2011*; *Amin et al., 2013*; *Pavlov et al., 2007*), hence we wanted to directly test the role of these actomyosin components in sea urchin skeletogenesis. To directly inhibit F-actin polymerization we used Latrunculin-A (Lat-A), that prevents the polymerization of actin monomers (*Yarmola et al., 2000*) and was shown to be effective in sea urchin embryos (*Schatten et al., 1986*). To directly inhibit actomyosin contractility we used Blebbistatin (Blebb), that interferes with MyoII ATPase activity, prevents actin-myosin interactions (*Kovács et al., 2004*), and was shown to be effective in sea urchin embryos (*Moorhouse et al., 2015*). To prevent interference with the early cell divisions where the actomyosin network plays a critical role (*Mabuchi, 1994*; *Schroeder, 1987*; *Wong et al., 1997*), we added the inhibitors, individually or together, before

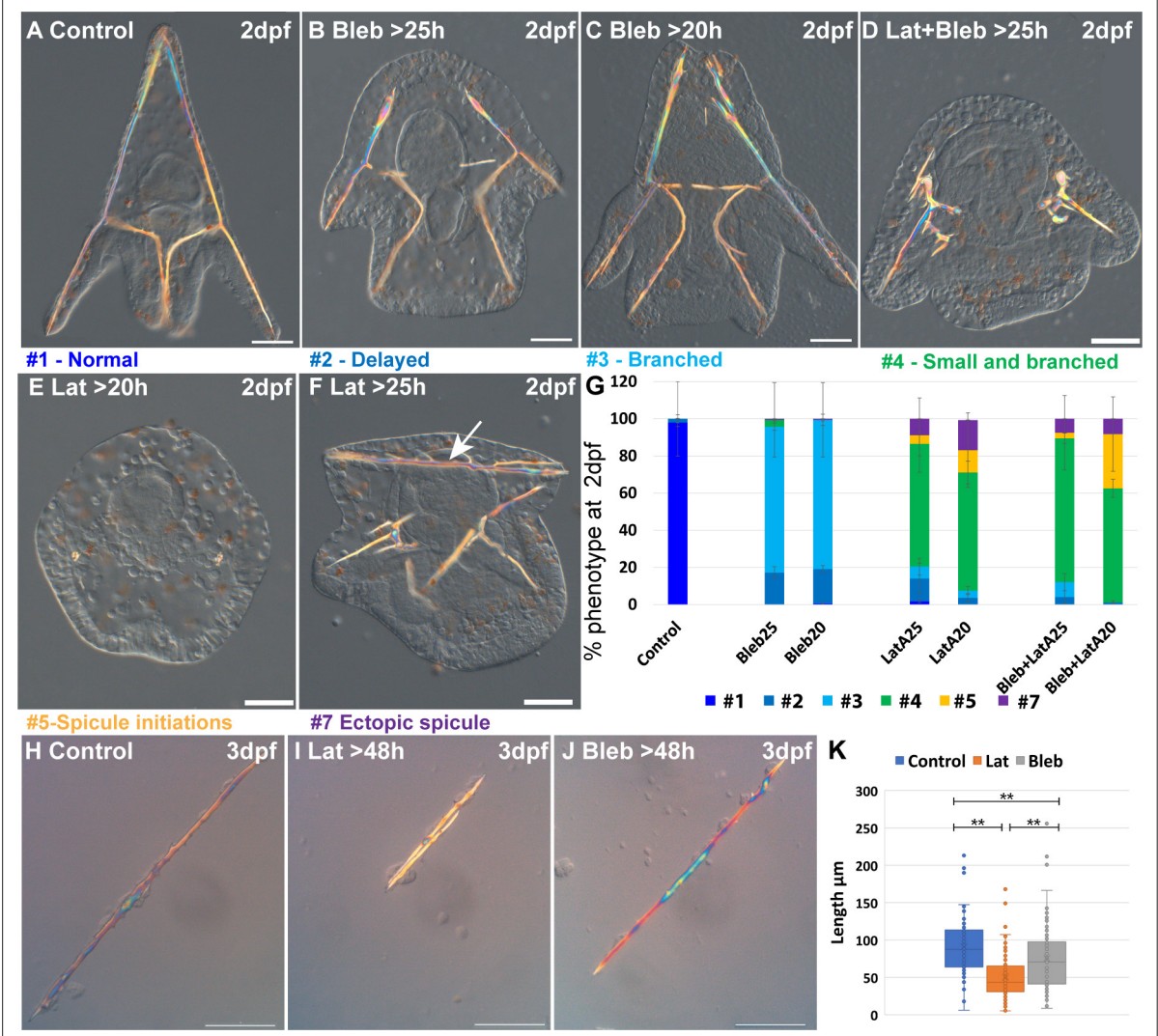

**Figure 4.** Actin polymerization and myosin activation affect skeletal growth and branching. (A–F) representative embryos showing the effect of actomyosin perturbations at 2df. (**A**) Control embryo (**B**) embryo treated with 2 µM Blebbistatin >25 hpf, (**C**) embryo treated with 2 µM Blebbistatin >20 hpf, (**D**) embryo treated with 2 nM Latrunculin-A and 1.5 µM Blebbistatin >25 hpf, (**E**) embryo treated with 2 nM Latrunculin-A >20 hpf, (**F**) embryo treated with 2 nM Latrunculin-A >25 hpf, arrow pointing to the additional spicule rod. (**G**) Statistics of Latrunculin-A and Blebbistatin phenotypes, color code of phenotype is indicated in the representative images. Error bars indicate standard deviation. All treatments were conducted in at least three biological replicates and the exact number of replicates and scored embryos are provided in *Supplementary file 1d*. (**H–J**) Representative spicules recorded at 72hpf from (**H**) control skeletogenic cell culture, (**I**) skeletogenic cell culture were 2 nM Latrunculin-A was added at 48hpf and (**J**) skeletogenic cell culture were 2 µM Blebbistatin was added at 48hpf. (**K**) Quantification of spicule length in the different treatments at 72hpf **p<0.001, Kruskal-Wallis non-parametric test. Results are based on three biological repeats for each treatment. Scale bars are 50 µm.

The online version of this article includes the following source data for figure 4:

**Source data 1.** Phenotypes of Blebbistatin and Latrunculin-A (Lat-A) experiments in whole embryos presented in *Figure 4G*.

**Source data 2.** Measurements of spicule length in skeletogenic cell cultures in control, of Blebbistatin and Latrunculin-A (Lat-A), presented in *Figure 4K*.

or after spicule formation (20hpf and 25hpf, see methods for details). We tested the resulting phenotypes at 2dpf (*Figure 4A–G*).

Our results indicate that F-actin polymerization and to a lesser extent, actomyosin contractility, are essential for normal skeletal growth and inhibition of ectopic branching. The majority of the embryos where MyoII activation was inhibited show ectopic branching mostly at the tips of the rods, and a minor delay in skeletal growth (*Figure 4B, C and G*). These phenotypes are quite similar to those observed in the addition of 10 µM of ROCK inhibitor at 25hpf (*Figure 2H and K*). The majority of the

embryos where F-actin formation was inhibited show a significant reduction of skeletal growth and severe ectopic skeletal branching (*Figure 4E and F* and similar phenotypes to *Figure 4D*, statistics in *Figure 4G*). These phenotypes are quite similar to those observed in 10 µM continuous inhibition of ROCK and the addition of 30 µM of ROCK inhibitor at 20hpf (*Figure 2H and K*). In some embryos, additional spicules form (*Figure 4F and G*). The co-inhibition of F-actin polymerization and MyoII activity results in skeletogenic phenotypes that resemble, but are slightly stronger, than those of the inhibition of F-actin alone (*Figure 4D and G*). Thus, the phenotypes resulting from the inhibition of MyoII activation resemble those of the late inhibition of ROCK at low inhibitor concertation. The phenotypes resulting from the inhibition of actin polymerization resemble those of the continuous inhibition of ROCK in low inhibitor concertation and the late inhibition of ROCK at higher inhibitor concentrations, yet skeletal loss is only observed under ROCK continuous inhibition.

To test whether actin polymerization and myosin activity specifically in the skeletogenic cells underlies the observed skeletogenic phenotypes, we inhibit both in an isolated skeletogenic cell culture (*Figure 4H–K*). We added the inhibitors after the spicule formed (>48 hpf) and observed the phenotypes at 3dpf. Both inhibitors reduced skeletal growth, with Lantruculin-A having a significantly stronger effect than Blebbistatin (*Figure 4K*), in agreement with the stronger skeletogenic phenotypes of Lantruculin-A in whole embryos. However, differently than in whole embryos, skeletal branching in both Lantruculin-A and Blebbistatin is not affected and is similar to the observed branching in control cultures. The absence of the branching phenotype in the skeletogenic cell culture could be due to the increased rigidity of the substrate that could compensate for the reduced actomyosin activity. Another option is that the branching phenotype in whole embryos is due to the reduction of the activity of the actomyosin network in non-skeletogenic cells. Nevertheless, these results indicate that normal skeletal growth depends primarily on F-actin polymerization and to a lesser extent, on myosin contractility in the skeletogenic cells.

## ROCK activity is required for F-actin organization around the forming spicule

The similarity between the skeletogenic phenotypes under ROCK inhibition and the direct perturbations of the actomyosin network led us to test the effect of ROCK inhibition on F-actin organization and MyoII activity. To accomplish this we used phalloidin, a fluorophore that binds F-actin specifically, an antibody that detects phosphorylated MyoII (MyoIIP, *Winter et al., 2021*) and the skeletogenic marker, 6a9 (*Figure 5*). In control embryos, F-actin is detected around the tri-radiate calcite spicules at 27hpf (green arrow in *Figure 5B*, in agreement with *Winter et al., 2021*; *Stepicheva et al., 2017*). The pseudopodia cable that connects the skeletogenic cells shows a lower F-actin signal compared to the signal around the spicule (blue line in *Figure 5B, C and E* blue arrows in *Figure 5G, H and J*). MyoIIP signal is not enriched in the skeletogenic cells nor in the spicule (*Figure 5D I*). Under continuous ROCK inhibition, the pseudopodia that connects the skeletogenic cells still forms (*Figure 5R*), and an enhanced F-actin signal is still observed in the lateral skeletogenic cell clusters (green arrow in *Figure 5L*); but unlike control embryos, the spicule cavity is not formed and F-actin is not organized around it (*Figure 5L*). MyoIIP signal seems unchanged by ROCK inhibition at this time (*Figure 5N and S*), in agreement with the weak and late phenotype of MyoII inhibition (*Figure 4*). Thus, the pseudopodia cable forms but F-actin organization around the spicules does not occur under continuous ROCK inhibition, however, it is not clear if the effect on F-actin organization is direct or due to the absence of spicule in this condition.

## F-actin is enriched at the tips of the elongating spicules independently of ROCK activity

To assess the role of ROCK in actomyosin organization during skeletal elongation we compared F-actin and myoIIP signals between control embryos and embryos where ROCK inhibitor was added at 25hpf (*Figure 6*). At the prism stage, we did not detect a clear difference in MyoIIP signal in the skeletogenic cells, between control and ROCK-inhibited embryos (*Figure 6D I*). At this time, F-actin is enriched at the tips of the spicules in both control and ROCK-inhibited embryos (white arrowheads in *Figure 6B and G*) as well as in ROCK morphants (green arrowheads in *Figure 2—figure supplement 1K, W*). In both control and ROCK-inhibited embryos, the F-actin signal was markedly higher in the regions of the pseudopodia cable where the spicule cavity had formed, compared to the regions where the

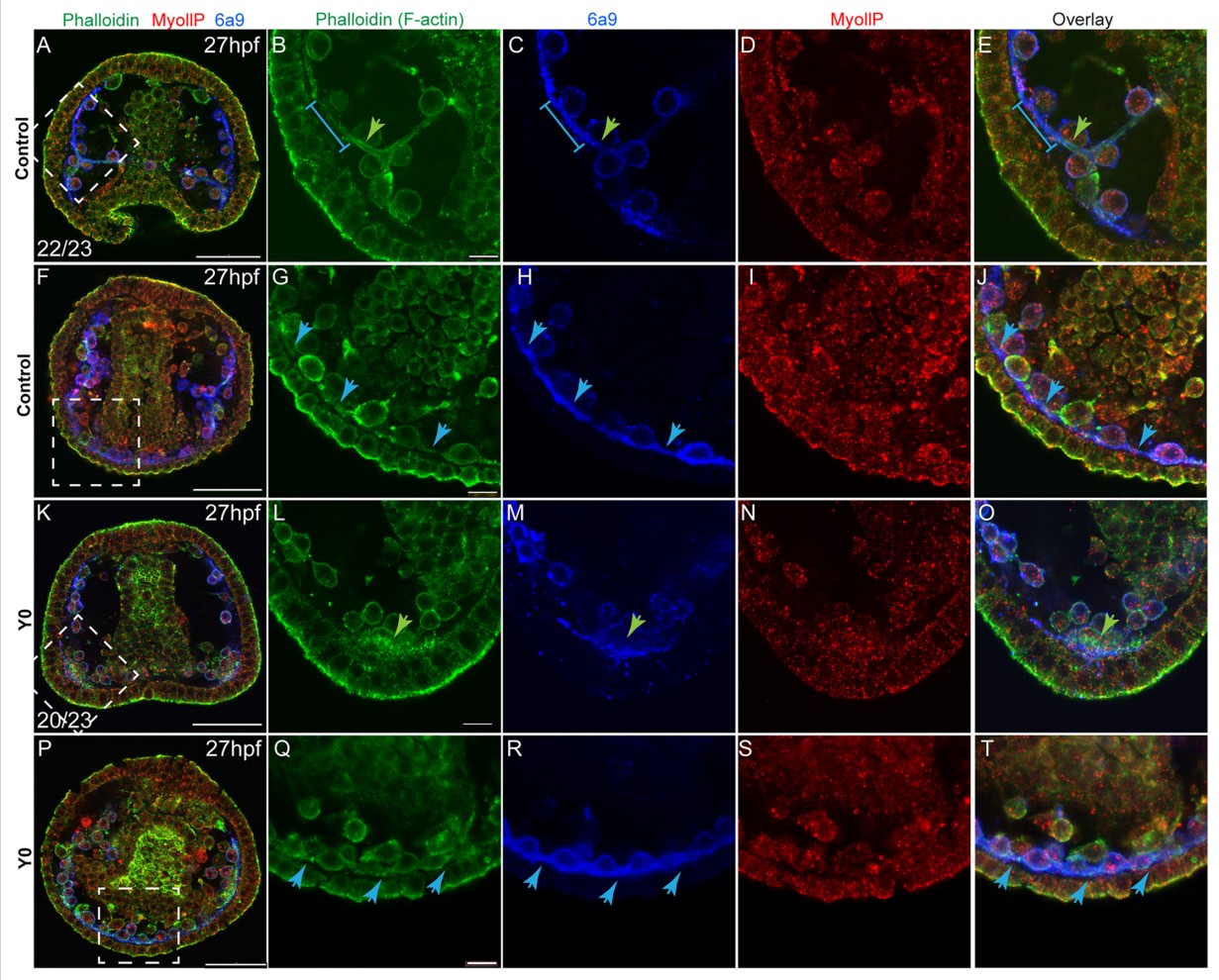

**Figure 5.** Rho-associated coiled-coil kinase (ROCK) inhibition effect on F-actin organization and MyoII activity at 27hpf. Representative images showing normal embryos (**A–J**) and embryos treated with ROCK inhibitor from fertilization (K-T, 80 µM), at 27hpf. Phalloidin (green) was used to stain F-actin, MyoIIP (red) was used to stain phosphorylated myosinII and 6a9 (blue) was used to mark the skeletogenic cells. Right panels show enlargements of the rectangle sections marked in the left most panels. Green arrows point to regions that show enriched F-actin signal, blue arrows point to the pseudopodia cable. The blue line in B, C, and E marks a region of the pseudopodia cable that is stained by 6a9 but has a low F-actin signal. The experiments were done in three biological replicates, the numbers at the bottom left of (**A, K**) indicate the number of embryos that show this phenotype out of all embryos scored. Scale bar in A, F, K, and P is 50 µm, and in B, I, N, and S is 10 µm.

spicule cavity had not yet formed (blue arrows in *Figure 6B–E and G–J*). We quantified the F-actin signal at the tips compared to the back in control and ROCK-inhibited embryos (*Figure 6K and L*, see methods for details). The phalloidin signal is on average threefold stronger at the tips of the spicules compared to the back, in both control and ROCK inhibition, indicating that F-actin enrichment at the tips is significant and independent of ROCK activity.

At the pluteus stage, F-actin is enriched at the tips of the spicules in both control and ROCK-inhibited embryos (white arrowheads in *Figure 6N–Q*)(*Figure 6*). Some non-skeletogenic cells that are enriched with MyoIIP signal are detected at this time (red arrowheads in *Figure 6N–Q*). Together these data demonstrate that F-actin coats the spicule cavity and is enriched at the tip of the rods, independently of ROCK activity.

We tested the effect of ROCK inhibition on F-actin in skeletogenic cell cultures. F-actin is detected at the tips, in both control cultures and cultures where ROCK was inhibited after spicule formation (*Figure 6W X*). Yet, under ROCK inhibition, branching is enhanced and F-actin is enriched at the splitting tips of the spicule rods (Fig, 6 X). The enrichment of F-actin at the splitting tips of the spicule under ROCK inhibition in skeletogenic cell culture resembles the calcein staining at the two tips in the

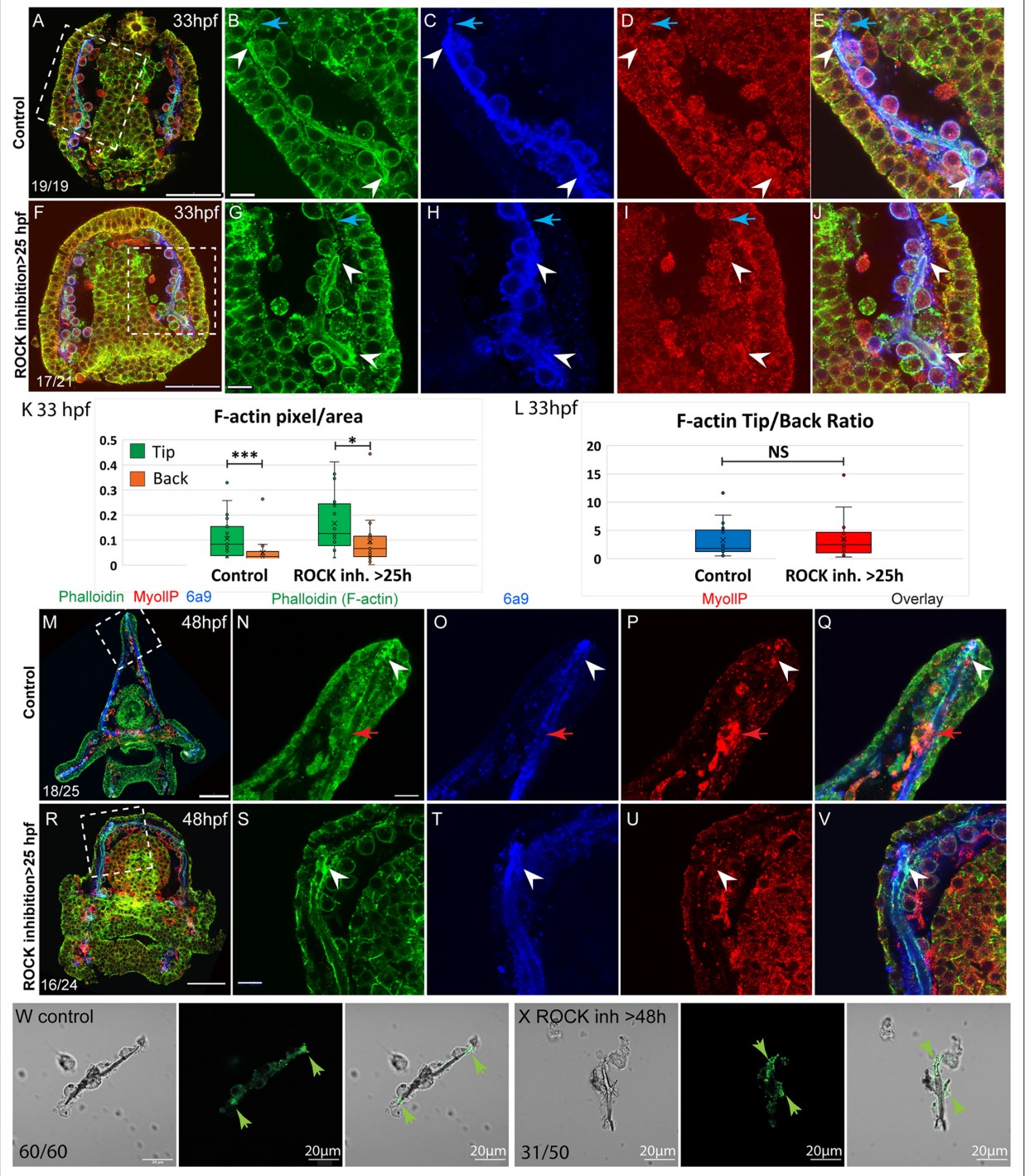

**Figure 6.** F-actin filaments are enriched at the tips of the spicules and the tip-to-back ratio is unaffected by ROCK inhibition. (**A–J**) Representative images at 33hpf, showing normal embryo (**A–E**) and embryos treated with 30 µM Rho-associated coiled-coil kinase (ROCK) inhibitor from the gastrula stage >25 hpf, (**F–J**). Embryos are stained with Phalloidin (green), MyoIIP antibody (red), and 6a9 (blue). (**B–E, G–J**) enlargement sections of the spicule area marked with rectangle in A and F. White arrowheads point to the enriched F-actin signal at the tips. Blue arrows point to the region of the pseudopodia cable that is not filled with the spicule cavity. (**K, L**) quantification of the tip-to-back F-actin signal (number of green pixels per area) at 33hpf in control embryos and ROCK inhibition >25 hpf. Each box plot shows the average (**x**), median (middle line), the first and the third quartiles, and all the points measured. Asterisks indicate statistical significance in paired t-test, where * is p<0.05 and *** is p<0.0005, NS- not significant. (**M–V**) similar experiments to (**A–J**), at 48hpf, 40 µM of ROCK inhibitor added at 25hpf. Red arrows point to non-skeletogenic cells enriched with MyoIIP. The experiments were repeated in three biological replicates and the numbers at the bottom left of (**A, F, M, R**) indicate the number of embryos that show this phenotype out of all embryos scored. Scale bar in A, F, M, R is 50 µm and in B, G, N, S, is 10 µm. (**W, X**) representative spicules out of three

*Figure 6 continued on next page*

Figure 6 continued

biological replicates from control skeletogenic cell culture (W, n=60), and skeletogenic cell treated with 30 μM ROCK inhibitor added at 48hpf and recorded at 72hpf (X, n=52). Left panel is phase image, middle panel is phalloidin staining and right panel shows the overlay. Green arrows point to the enhanced F-actin signal at the tips. Scale bar is 20 μm.

The online version of this article includes the following source data for figure 6:

**Source data 1.** Measurements of F-actin signal in the tips vs. the back of the spicules in control and under Rho-associated coiled-coil kinase (ROCK) inhibition, presented in *Figure 6K and L*.

embryo in this condition (*Figure 3I*). These observations further support the role of ROCK activity in regulating tip-dominance during sea urchin skeletogenesis.

## ROCK activity is essential for normal skeletogenic gene expression

The role of ROCK in regulating gene expression during vertebrates' chondrocytes, osteoblasts, and odontoblasts differentiation (*Strzelecka-Kiliszek et al., 2017*; *Qin et al., 2020*; *Huang et al., 2018*) intrigued us to study the effect of ROCK inhibition on skeletogenic gene expression in the sea urchin embryo. ROCK continuous inhibition significantly downregulates the expression of multiple skeletogenic genes, including the cytoskeleton remodeling gene, *rhogap24l/2*, the biomineralization genes, *caral7*, *scl26a5*, and *SM30* at 27hpf and 2dpf (*Figure 7A*, *Morgulis et al., 2019*). ROCK inhibition did not affect the expression level of VEGFR and ROCK itself at these times (*Figure 7A*). Washing the inhibitor after 25hpf or adding the inhibitor at 25hpf had a weaker effect, but still led to

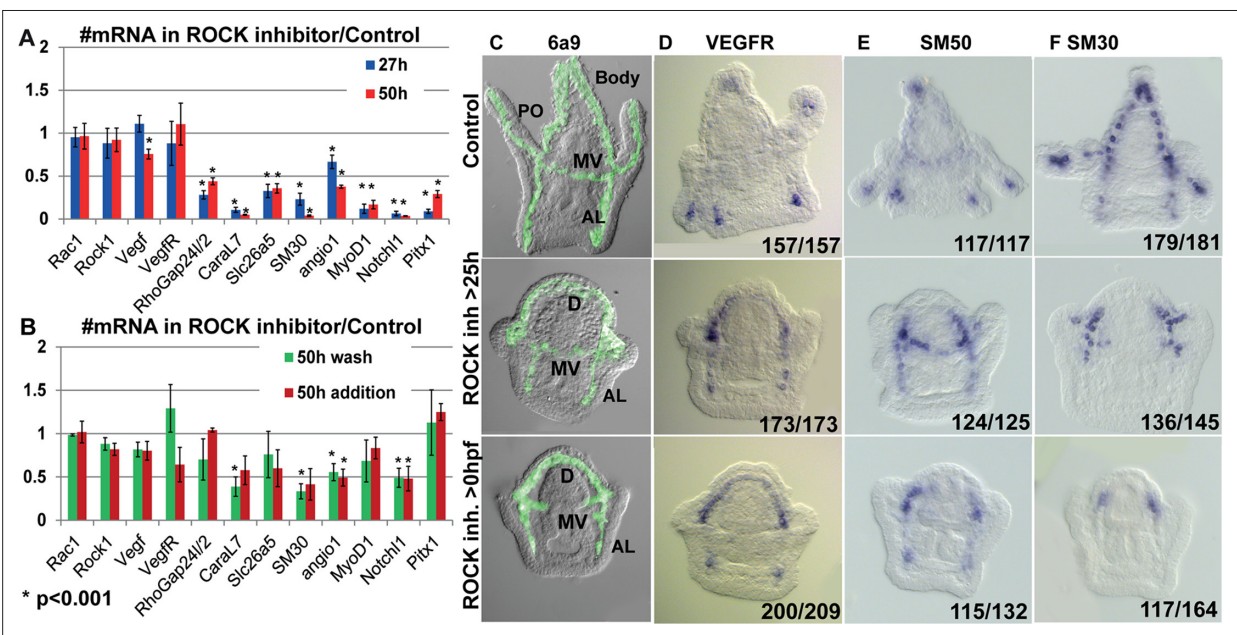

**Figure 7.** Rho-associated coiled-coil kinase (ROCK) activity is essential for normal gene expression in the skeletogenic cells. (**A, B**) The effect of 80 μM ROCK inhibition on gene expression. (**A**) Continuous ROCK inhibition at 27hpf and 50 hpf (n=4). (**B**) Addition of ROCK inhibitor at 25hpf and the wash of ROCK inhibitor at 25hpf, measured at 50hpf (n=3). Asterisks indicate p<0.001, one-tailed z-test. Error bars show standard deviation. (**C–F**) Representative images of control embryos (top panels), embryos where ROCK inhibitor was added at 25hpf (middle panels), and embryos that were exposed to continuous ROCK inhibition (bottom panels), at the pluteus stage (~48 hpf). (**C**) skeletogenic cell marker, 6a9. MV, midventral; AL, anterolateral, and PO, Post-oral rods, D, dorsal skeletogenic chain. (**D–F**) spatial expression of skeletogenic genes. Gene names are indicated at the top of each panel. Numbers at the bottom of each image indicate the number of embryos that show this phenotype (left) out of all embryos scored (right), conducted in at least three independent biological replicates.

The online version of this article includes the following source data and figure supplement(s) for figure 7:

**Source data 1.** Quantitative polymerase chain reaction (QPCR) measurements of gene expression in control and under ROCK inhibition presented in *Figure 7A and B*.

**Figure supplement 1.** Rho-associated coiled-coil kinase (ROCK) activity is essential for normal gene expression in the skeletogenic cells.

downregulation of *caral7*, *SM30*, *angio1,* and *notch1* (*Figure 7B*). Thus, ROCK activity is required for the normal expression level of multiple skeletogenic genes during sea urchin skeletogenesis.

Both continuous and late ROCK inhibition strongly affect the spatial expression of key regulatory and biomineralization genes that are normally expressed differentially within the skeletogenic lineage at the pluteus stage (*Figure 7D–F, Figure 7—figure supplement 1*). These include the VEGF receptor (VEGFR), the biomineralization genes *SM50* and *SM30,* skeletogenic transcription factors, Ets1, Alx1, and Erg1 that are essential for skeletogenic cell specification and biomineralization (*Oliveri et al., 2008*; *Khor and Ettensohn, 2020*), and the transcription factors Hex and MyoD1. In normal plutei, these genes are differentially expressed within skeletogenic lineage, but in both continuous and late ROCK inhibition the expression of these genes becomes localized to the two lateral skeletogenic cell clusters and their vicinity (*Figure 7D–F, Figure 7—figure supplement 1*). The lateral skeletogenic cell clusters are the cells where these genes are expressed in the gastrula stage (24-27hpf) and where spicule formation begins in normal embryos (*Morgulis et al., 2019*; *Tarsis et al., 2022*). The expression of VEGFR and SM50 under ROCK inhibition is more expanded than the other tested genes and is also observed at the anterolateral rods (*Figure 7D and E*). The VEGF ligand that is normally expressed in the ectodermal cells near the four tips of the post-oral and anterolateral rods (*Tarsis et al., 2022*), is expressed at four ectodermal domains near the lateral skeletogenic cell clusters and near the end of the anterolateral chain, under ROCK inhibition (*Figure 7—figure supplement 1*). Our data shows that under ROCK inhibition, the expression of key skeletogenic genes remains localized to the area near the lateral skeletogenic cell clusters (*Figure 7D–F, Figure 7—figure supplement 1*), despite the proper formation of the dorsal, anterolateral and mid-ventral chains (*Figure 7C*). Overall, ROCK activity is essential for normal expression level and spatial gene expression in the skeletogenic lineage.

## Discussion

Biomineralization is a complex morphogenetic process, encoded in the species genome and executed by the GRNs and the cellular machinery. Here, we used the sea urchin larval skeletogenesis to

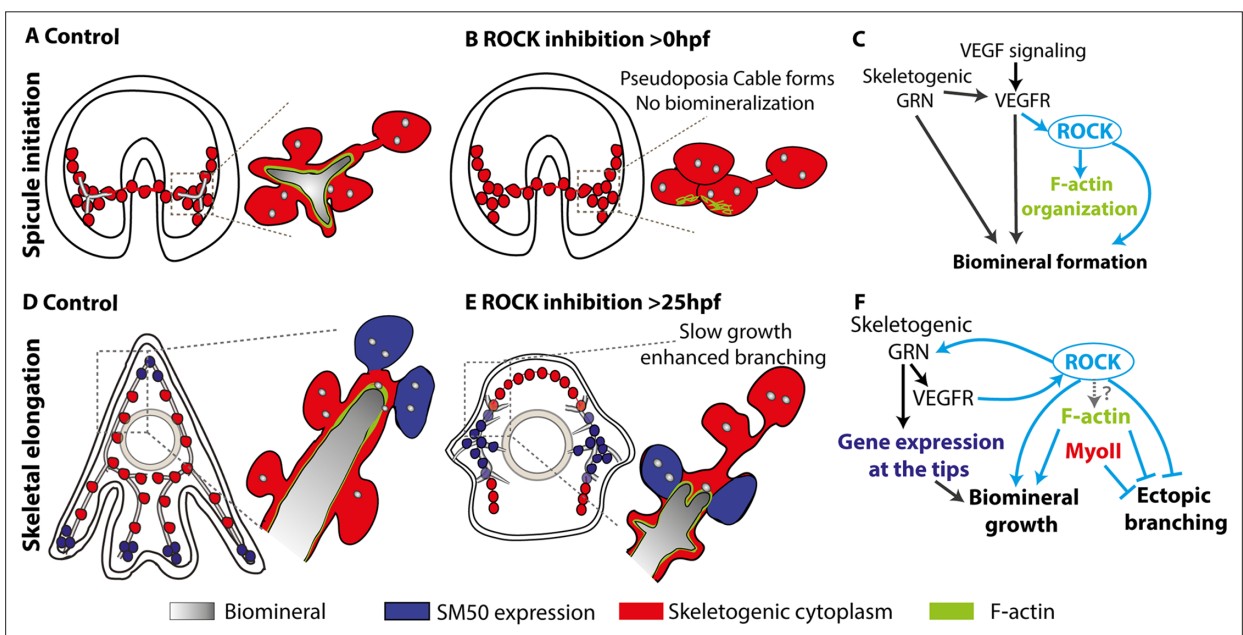

**Figure 8.** The role of Rho-associated coiled-coil kinase (ROCK) and the actomyosin network in sea urchin skeletogenesis, summary. (**A**) The spicule forms at about 24hpf in *P. lividus* and at 27hpf, the triradiate spicule is coated with F-actin. (**B**) ROCK activity is required for spicule initiation and for F-actin organization around the spicule. (**C**) model of the functional links between the skeletogenic gene regulatory network (GRN), ROCK, F-actin, and their skeletogenic outcomes, during spicule initiation (~24–27 hpf in *P. lividus*). Cian arrows indicate links discovered in this work. (**D**) During skeletal elongation, F-actin is detected around the spicule cavity and is enriched at the tips of the rods. The expression of SM50 and some other skeletogenic genes, is localized to the tips of the rods. (**E**) under the addition of ROCK inhibitor, skeletal growth rate is reduced, and ectopic spicule branching is observed. The expression of skeletogenic genes is localized to the vicinity of the growing rods. (**F**) the functional links between the skeletogenic GRN, ROCK, F-actin, and MyoII and their skeletogenic outcomes during skeletal elongation. Cian arrows indicate links discovered in this work.

investigate the role of ROCK and the actomyosin network in biomineral formation and in regulating gene expression, as we summarize in *Figure 8*. ROCK expression is enriched in the skeletogenic cells, partially depending on VEGF signaling (*Figures 1 and 8C*). ROCK activity in the skeletogenic cells is necessary for spicule formation, skeletal elongation, and prevention of ectopic skeletal branching (*Figures 2 and 3*, *Figure 2—figure supplements 2 and 3*). Direct inhibition of F-actin polymerization results in similar but not identical skeletogenic phenotypes (*Figure 4*). F-actin is organized around the spicule cavity and enriched at the spicule tips (*Figures 5 and 6*). ROCK activity feeds back into the skeletogenic GRN and affects regulatory and biomineralization gene expression (*Figure 7*). Below we discuss our findings and their implications on the biological control and evolution of biomineralization.

The first step in generating the sea urchin spicules is the construction of the spicule cavity where the mineral is engulfed in a membrane coated with F-actin (*Figure 8A*). ROCK activity is necessary for the spicule initiation and for the organization of F-actin around the spicule cavity (*Figures 2, 5 and 8B*). However, since the spicule doesn't form under ROCK inhibition, it is hard to conclude if the absence of F-actin organization around the spicule is due to ROCK regulation of actin polymerization or due to the absence of spicule in this condition. Relatedly, a direct perturbation of F-actin polymerization strongly disrupts spicule growth and shape, but not spicule formation, indicating that F-actin is not essential for spicule initiation (*Figure 4*). We, therefore, propose that ROCK activity is necessary for spicule initiation, but not through its' direct regulation of the actomyosin network, but though its other targets, yet to be identified (*Figure 8C*).

After the spicule cavity has formed, the spicules elongate inside the pseudopodia chord by mineral deposition at the tips of the spicule rods, which depends on ROCK activity, on F-actin polymerization, and to a lesser extent, on MyoII contractility (*Figures 2–4 and 8D–F*). F-actin is detected along the spicules and is significantly enriched at the tips of the elongating spicules, independently of ROCK activity (*Figures 5, 6 and 8D, E*; *Miklavc and Frick, 2020*). Under ROCK inhibition and under the inhibition of F-actin polymerization, ectopic branching is observed at the tips and in the back (*Figure 2H, J, O*; *Figure 3A, I*; *Figure 4*; *Figure 8E*). MyoII inhibition results in ectopic branching at the tips of the rods (*Figure 4*), that is similar to the effect of late ROCK inhibition at low concertation (*Figure 2H and K*). These correlative similarities between ROCK and the actomyosin perturbations lead us to the following speculations: the low dosage of late ROCK inhibition is perturbing mostly ROCK activation of MyoII contractility while the higher dosage affects factors that control actin polymerization (*Figure 8F*). Further studies in higher temporal and spatial resolution of MyoIIP activity and F-actin structures in control and under ROCK inhibition will enable us to test this.

The participation of ROCK, F-actin, and MyoII in polarized growth and vesicle exocytosis has been observed in both animals and plants (*Miklavc and Frick, 2020*; *Bibeau et al., 2018*). F-actin is used for short-range myosin trafficking of vesicles toward the secretion site (*Miklavc and Frick, 2020*; *Bond et al., 2011*). Once the secretory vesicle fuses with the membrane, it is coated with F-actin that stabilizes the vesicle and enables MyoII binding and ROCK-regulated contraction, necessary for content secretion (*Segal et al., 2018*; *Rousso et al., 2016*). In plant cells, F-actin accumulates at the growing tip of the cell and assists in vesicle exocytosis necessary for polarized tip-growth (*Bibeau et al., 2018*). ROCK, F-actin, and MyoII could be regulating the localized exocytosis of mineral-bearing vesicles during sea urchin skeletogenesis. The reduction of spicule growth rate and the enhanced branching could be due to impaired and misplaced vesicle trafficking and deposition under ROCK and F-actin inhibition. Detailed investigation of the kinetics and deposition of mineral-bearing vesicles and the role of the actomyosin network in these processes will illuminate the regulation of vesicle exocytosis during biomineralization.

ROCK activity is necessary for correct spatiotemporal expression of regulatory and biomineralization genes (*Figure 7*). ROCK inhibition downregulates the expression level of skeletogenic genes (*Figure 7A*) and prevents the spatial progression of key skeletogenic genes toward the edges of the skeletal rods (*Figures 7 and 8E*, *Figure 7—figure supplement 1*). This implies that ROCK activity provides regulatory cues to the skeletogenic GRN required for the dynamic progression of gene expression within the skeletogenic lineage. In other systems, ROCK activity was shown to regulate gene expression through various intracellular pathways (*Olson and Nordheim, 2010*; *Knipe et al., 2015*; *Khatiwala et al., 2009*). Furthermore, ROCK has an important role in the mechano-sensing of matrix rigidity and transducing it into changes in gene expression (*Zhong et al., 2020*). Relatedly, stiff substrate activates ROCK in pre-osteoblastic cells in-vitro, which activates Erk that leads to osteogenic

differentiation and upregulation of osteogenic gene expression (*Khatiwala et al., 2009*). Sea urchin ROCK could be a part of the mechano-sensing mechanism that detects the high spicule stiffness and transduces it into gene expression, providing a necessary feedback between spicule elongation and the skeletogenic GRN.

Overall, our findings together with the role of the actomyosin network in biomineralization, from calcifying and silicifying single cells to vertebrates' bones and teeth (*Langer et al., 2010*; *Durak et al., 2017*; *Tyszka et al., 2019*; *Tesson and Hildebrand, 2010a*; *Tesson and Hildebrand, 2010b*; *Strzelecka-Kiliszek et al., 2017*; *Qin et al., 2020*; *Huang et al., 2018*; *Khatiwala et al., 2009*), suggest that this molecular machinery is a part of the common molecular tool-kit of biomineralization. Most likely, the actomyosin network was employed independently, downstream of distinct GRNs across Eukaryotes, to control mineral growth and morphogenesis.

# Methods

## Key resources table

| Reagent type (species) or resource | Designation | Source or reference | Identifiers | Additional information |
|---|---|---|---|---|
| Antibody | ROCK2 +ROCK1 antibody | Abcam | AB-ab45171 | IF (1:70) WB (1:300) |
| Sequence-based reagent | MASO | This paper | ROCK MASO1 | AGACATATTT GGAGCCGA [CAT]CCTG |
| Sequence-based reagent | MASO | This paper | ROCK MASO2 | TCTCTTG CGTTATAT TCCACTAAGT |
| Chemical compound, drug | Y27632 | Cayman chemical | 10005583 CAS Registry No. 129830-38-2 | 10–120 µM |
| Chemical compound, drug | Axitinib | Selleckchem, Houston, TX, USA | AG013736 | 150 nM |
| Chemical compound, drug | Latrunculin-A | Thermo Fisher Scientific | L12370 | 2 nM |
| Chemical compound, drug | Blebbistatin | Enco | 13013–5 | 1.5–2 µM |

## Animal and embryos

Adult *Paracentrotous lividus* were obtained from the Institute of Oceanographic and Limnological Research (IOLR) in Eilat, Israel. Eggs and sperm were obtained by injection of 0.5 M KCl solution into the adult sea urchin species. Embryos were cultured at 18 °C in 0.2µ filtered ASW.

## Imaging

Embryonic phenotypes and WMISH were imaged by Zeiss Axioimager 2. Fluorescent markers were imaged using a Nikon A1-R Confocal microscope. All images were aligned in Adobe photoshop and Illustrator.

## VEGFR inhibitor, Axitinib (AG013736) treatment

Axitinib (AG013736, Selleckchem, Houston, TX, USA) was applied as described in *Tarsis et al., 2022*.

## Western blot

Embryo cultures treated with 150 nM Axitinib, or DMSO as control, were collected at 22hpf, 27hpf, and 33hpf by centrifugation of 10,000 g for 2 min. We lysed the pellet in 150 µL lysis buffer (20 mM Tris-HCl, 150 mM NaCl, 1% Triton X-100, pH 8) with protease inhibitors (protease inhibitor cocktail w/o metal chelator; Sigma, P8340; 1:100) and phosphatase inhibitors (50 mM NaF, 1 mM Na3VO4, 1 mM Na4P2O7, 1 mM, βGP) as described in *Luo and Su, 2012*. 40 µg protein were loaded on 8% SDS-acrylamide gel, transferred to PVDF membranes for Western blot analysis. Anti-ROCK antibody (ab45171, Abcam) was used in a 1:300 dilution followed by an anti-rabbit HRP secondary antibody

diluted to 1:2500 (Jackson ImmunoResearch 111-035-003). For the loading control, the membrane was incubated with an anti-β-tubulin antibody (1:5000; Sigma) followed by an anti-mouse antibody 1:5000 (Jackson ImmunoResearch 115-035-003). Quantification of ROCK signal was done using Image studio lite vr. 5.2. ROCK signal was divided by the Tubulin signal and then the ratio between ROCK/Tubulin in control vs. VEGFR inhibition was calculated. The graph in *Figure 1—figure supplement 2* was generated based on three biological replicates (different sets of parents) at 27hpf and 33hpdf and four biological replicates at 22hpf.

## Immunostaining procedure

Phalloidin labeling, MyoIIP (p20-MyosinII), ROCK (anti ROCK2 +ROCK1 antibody [EP786Y], AB-ab45171, Abcam, 1:70), and 6a9 immunostaining were done similarly to *Winter et al., 2021*.

## Quantification of anti-ROCK and phalloidin signal

We used a graphical program to manually quantify the fluorescent signal from the spicule regions by identifying 'stained' pixels per selected area (*Winter et al., 2021*). The ratio of stained anti-ROCK to region total area was compared between ectodermal and skeletogenic cells, and between control and VEGFR-inhibited embryos at 22, 27, and 33hpf. The average ratios and z-test for significance difference from 1, are 22hpf control, ~1.3, z=0.0006; VEGFR inhibition ~1, z=0.5, 27hpf control ~1.8, z<10–11; VEGFR inhibition ~1.6, z<10–8, 33hpf control ~4.6, z<10–23; VEGFR inhibition ~1.5, z<10–5. For phalloidin analyses, embryos at 33hpf of control and ROCK inhibitor addition at 25hpf were used and for each image, two regions were selected: an area at the tip of the spicule and an area at the back of the spicule, about ~10 microns apart. The ratio of stained phalloidin to region total area was compared between tip and back, in control and ROCK-inhibited embryos. A paired t-test was used to compare the differences in phalloidin signal between the tip and back of the spicules across all groups and between control and ROCK-inhibited embryos.

## ROCK MASOs injections

Microinjections were conducted similarly to *Tarsis et al., 2022*. The eggs were injected with an injection solution containing 0.12 M KCl, 0.5 µg/µl Rhodamine Dextran, and 900 µM of MASOs. Translation MASOs were designed and synthesized by Gene Tools, Philomath, OR, according to ROCK sequence (*Marlétaz et al., 2023*). The sequence of ROCK MASO-1 (5'-AGACATATTTGGAGCCGA[CAT]CCTG-3') matches the start of the translation region and ROCK MASO-2 (5'-TCTCTTGCGTTATATTCCAC TAAGT-3') matches the 5' region upstream of the start of translation (*Marlétaz et al., 2023*). Control embryos were injected in the same concentration with Random MASO. Injected embryos were cultured at 18 °C, imaged, and scored at 2dpf.

## ROCK inhibitor (Y27632) treatment

Y27632 stock (10005583 CAS Registry No. 129830-38-2, Cayman chemical), was prepared by reconstituting the chemical in DMSO. The embryos were treated with Y27632 at a final concentration between 30–80 µM, as mentioned in the results. Throughout the paper, control embryos were cultured in DMSO at the same volume as Y27632 solution and no more than 0.1% (v/v).

## ROCK inhibition in isolated skeletogenic cell culture

Skeletogenic cell culture was performed as described in *Moreno et al., 2019* with minor changes. Isolated skeletogenic cells were cultured in CultureSlides (four chambers polystyrene vessel tissue culture treated glass slide REF 354104) with ASW (Red Sea Fish LTD) containing Gentamicin and Penicillin-Streptomycin (GPS) at 18 °C. Media containing 4% (v/v) horse serum (Sigma-Aldrich H1270) in ASW +GPS was added to the culture. ROCK inhibitor, Y27632, was added to the cell culture at 16hpf (before spicule initiation) or 48hpf (after spicule initiation), and images were taken at 72hpf. Each treatment was conducted in three biological repeats except for the >48 hpf addition of 120 µM which was done in two biological repeats.

## Quantification of skeletal length and statistical analysis

Skeletal length measurement was done as described in *Tarsis et al., 2022*. The measurements were repeated for three biological repeats for control, ROCK inhibition with 30 µM and 80 µM, Latrunculin-A

2 nM, Blebbistatin 2 µM, and two biological repeats for 120 µM. In ROCK inhibition experiments, a total of 275 skeletons were measured for control, 105 skeletons for 30 µM, 122 skeletons for 80 µM, and 80 for 120 µM. In Latrunculin-A and Blebbistatin experiments, a total of 116 skeletons were measured for control, 149 for Latrunculin-A, and 107 for Blebbistatin. The data were analyzed in Excel and the statistical analysis was performed using Kruskal-Wallis non-parametric test in Excel as described in *Siegel and Castellan, 1988* and SPSS statistics 27.

## Spicule preparation for micro-computed tomography (SR-µCT) measurement

Control embryos and embryos where 40 µM Y27632 was added at 25hpf, were collected by centrifugation of 50 mL Falcon tubes at 2000 rpm, at 48 hr and 72hpf. Every tube was washed in 10 mL cold distilled water (DW), transferred into an epi-tube, and washed three times in 1 mL cold DW (incubation at 4 °C between every wash, centrifuge at 2000 rpm in room temperature for 2 min). Then the skeletons were washed three times in 3% NaOCl followed by three washes in DW, one wash in 70% EtOH, and one wash in Acetone. Skeletal spicules were dried at room temperature and stored at –20 °C.

## Synchrotron and lab-based SR-µCT

Dry and loose skeletal spicules prepared as described above, were used for the SR-µCT analysis. We analyzed 14 sets of spicules from three (control) and four (ROCK inhibited) pairs of parents, divided into four groups: 2dpf, control, n=44; 2dpf ROCK inhibited, n=93; 3dpf control, n=51; 3dpf ROCK inhibited, n=93, see *Supplementary file 1a*. Spicules were glued to sharpened toothpick tips, either in bulk or individually when picked by hand by means of a brush bristle. Each toothpick loaded with spicules, was mounted on a 3 ml vial twist-off lid and for transportation securely stored in the vial (*Figure 2—figure supplement 3A, B*). For each set, 2–3 µCT-samples were prepared, which allowed the acquisition of statistically relevant sample sizes (40+ spicules/group, *Supplementary file 1a*). Tomographic data sets were acquired using the following scanning parameters; for the lab-based µCT (RX Solutions, Chavanod, France) 90kV, 143-160µA, 1440 images, average frames 10, frame rates 0.4–1, voxel sizes: 0.45–59 µm; and for the synchrotron radiation µCT (Anatomix, Synchrotron SOLEIL, Gif-sur-Yvette, France) 2048 images, angle between projections 0.09°, voxel size: 0.65 µm. Tomographic data reconstruction was performed with commercial (RX Solutions, Chavanod, France) and PyHST2 software (A. Mirone et al., Nucl. Instrum. Meth. B 324 (2014) 41–48, doi:10.1016 /j.nimb.2013.09.030) at the MPICI and at Anatomix, SOLEIL, respectively.

## Three-dimensional (3D) data analysis

Data visualization, pre-processing including cropping and image type conversion to 16bit (of the Sr- µCT data), and user-augmented segmentation of intact spicules were performed in Amira using off-the-shelf modules (*Figure 2—figure supplement 3C–E*, *Stalling et al., 2005*). Volume and area measurements were performed on the segmented spicules, i.e., the label-fields using Amira's material.statistics module (Volume3D and Area3D). Average spicule thickness and total length measurements were performed with python code, utilizing available python libraries (e.g. skimage, skan, scipy). The code was applied to 3D-tif files created from Amira's label-fields, which contained the segmented spicules. For length measurements, the spicules were skeletonized and the chord lengths of the branches forming a spicule-skeleton were summed up to obtain the total spicule length. For thickness measurements, we performed Euclidean distance transforms for each spicule using scipy's ndimage.morphology.distance_transform_edt package, and calculated the average spicule thickness based on the distance measurements obtained at each voxel from the spicule-skeleton to the nearest spicule surface. Quality control of our measurements was performed with open-source software, Fiji, and available plugins (e.g. BoneJ, FiberJ, DiameterJ).

## Calcein staining

A 2 mg/ml stock solution of blue calcein (M1255, Sigma) and a 2 mg/ml stock solution of green calcein (C0875, Sigma) were prepared by dissolving the chemicals in distilled water. The working solution of 250 µg/ml and 150 µg/ml, respectively, was prepared by diluting the stock solution in artificial sea water. Blue calcein was added to the embryo culture at 29hpf for 1 hr and then washed. At 44hpf the green calcein was added for 1 hr and washed. For FM4-64 staining A 100 µg/ml stock solution

of FM4-64 (T13320, Life technologies, OR, USA) was prepared by dissolving the chemical in distilled water. The working solution of 2.5 µg/ml was prepared by diluting the stock solution in artificial sea water. Embryos were immersed in the working solution about 10 min before visualization.

## Latrunculin-A and Blebbistatin treatments

1 mM of Lat-A stock (L12370, Thermo Fisher Scientific) was prepared by reconstituting the chemical in DMSO. The embryos were treated with Lat-A at a final concentration of 2 nM. 1 mM of Blebbistatin stock (13013–5, Enco) was prepared by reconstituting the chemical in DMSO. The embryos were treated with Blebbistatin at a final concentration of 2 µM. For the double inhibition experiments, embryos were treated with Blebb at a final concentration of 1.5 µM, and with Lat-A at a final concentration of 2 nM.

## WMISH procedure

WMISH was performed as described in *Tarsis et al., 2022*. A list of the primers used to make the WMISH probes is available in *Tarsis et al., 2022*.

## cDNA preparation for QPCR experiments

For ROCK inhibition experiments total RNA was extracted from >1000 sea urchin embryos in each condition using RNeasy Mini Kit (*Beane et al., 2006*) from QIAGEN (#74104) according to the kits' protocol. DNase treatment on the column was done using RNase-Free DNase Set-Qiagen (*Beane et al., 2006*) (#79254). RNA was reverse-transcribed using the High Capacity cDNA RT kit, AB-4368814 (Applied Biosystems) following the manufacturer's instructions.

## Quantitative polymerase chain reaction (QPCR)

QPCR was carried out as described in *Tarsis et al., 2022*. Complete list of primer sequences used for the experiments in provided there.

## Data availability statement

The authors confirm that the data supporting the findings of this study are available within the article and its supplementary materials. The gene and protein sequences that support the findings of this study are available in the following public databases: *P. lividus* data is available at https://www.ncbi.nlm.nih.gov/bioproject/834074 (*Marlétaz et al., 2023*), *S. purpuratus* data is from echinobase, https://echinobase.org/echinobase/ (*Telmer et al., 2024*) and human sequences are from NCBI https://www.ncbi.nlm.nih.gov/datasets/gene/.

## Acknowledgements

We are grateful to SOLEIL for the provision of synchrotron radiation facilities and we would like to thank Dr. Mario Scheel, Dr. Timm Weitkamp, and Dr. Jonathan Perrin for assistance in using beamline ANATOMIX for nano and µCT. We thank the Bioimaging Unit, Faculty of Natural Sciences, University of Haifa for assistance with the use of confocal. We thank Charles Ettensohn for the generous gift of the 6a9 antibody produced in his lab. We thank David Ben-Ezra, Michael Kantorovitz, and Alvaro Israel for their help with sea urchin handling. We thank Haguy Wolfenson for illuminating discussions of the results and conceptual framework. We thank Nir Sapir and Tovah Nehrer for their advice and help with the statistical analysis of spicule length. We thank Aleksei Tabachnic for his help with microinjection. We thank Alvaro Israel and Omri Nahor for their help with sea urchin maintenance. We thank Yulia Kagan for her help with the western blot. Israel Science Foundation grant number 211/20 (SBD), and Zuckerman fellowship (MRW), TUD internal funding (YP). ANATOMIX is an Equipment of Excellence (EQUIPEX) funded by the Investments for the Future program of the French National Research Agency (ANR), project NanoimagesX, grant no. ANR-11-EQPX-0031. Access to ANATOMIX beamline is granted for proposal number 20200811 (YP).

# Additional information

## Funding

| Funder | Grant reference number | Author |
|---|---|---|
| Israel Science Foundation | 211/20 | Smadar Ben-Tabou de-Leon |
| ANATOMIX | 20200811 | Yael Politi |

The funders had no role in study design, data collection and interpretation, or the decision to submit the work for publication.

## Author contributions

Eman Hijaze, Conceptualization, Formal analysis, Validation, Investigation, Visualization, Methodology, Writing – original draft, Writing – review and editing; Tsvia Gildor, Conceptualization, Data curation, Formal analysis, Supervision, Investigation, Methodology, Project administration, Writing – review and editing; Ronald Seidel, Software, Formal analysis, Investigation, Visualization, Methodology, Writing – review and editing; Majed Layous, Formal analysis, Investigation, Visualization, Methodology, Writing – review and editing; Mark Winter, Software, Formal analysis, Methodology; Luca Bertinetti, Software, Supervision, Methodology; Yael Politi, Supervision, Investigation, Methodology, Writing – review and editing; Smadar Ben-Tabou de-Leon, Conceptualization, Formal analysis, Investigation, Supervision, Visualization, Writing – original draft, Writing – review and editing

## Author ORCIDs

Mark Winter ⓘ http://orcid.org/0000-0003-1180-1957
Smadar Ben-Tabou de-Leon ⓘ https://orcid.org/0000-0001-9497-4938

Reviewer #1 (Public Review): https://doi.org/10.7554/eLife.89080.4.sa1
Reviewer #2 (Public Review): https://doi.org/10.7554/eLife.89080.4.sa2
Author response https://doi.org/10.7554/eLife.89080.4.sa3

# Additional files

## Supplementary files

• Supplementary file 1. Additional statistical information for ROCK inhibition and micro-CT experiments. (a) Rho-associated coiled-coil kinase (ROCK) inhibition experimental details. The table provides the number of biological replicates and embryos scored for each experiment with ROCK inhibitor. (b) µCT statistics of control and ROCK inhibited spicules. The table provides the details of the number of spicules measured, average and standard deviation of measured length, thickness, volume, and surface area. (c) µCT statistical significance. The table provides the statistical significance between µCT measurements of control and ROCK-inhibited spicules at 2dpf and 3dpf, based on the parametric student's T-test. (d) Lantrunculin-A and Blebbistatin experimental details. The table provides the number of biological replicates and embryos scored for each experiment with Lantrunculin-A and Blebbistatin.

• MDAR checklist

## Data availability

The authors confirm that the data supporting the findings of this study are available within the article and its supplementary materials.The numerical data used to generate the figures is uploaded in the source data files of Figures 1, 2, 3, 4, 6 and 7.

The following previously published datasets were used:

| Author(s) | Year | Dataset title | Dataset URL | Database and Identifier |
|---|---|---|---|---|
| Marletaz F | 2023 | Comparative genomics of the sea urchin P. lividus highlights contrasting trends of genome and regulatory evolution in deuterostomes (common urchin) | https://www.ncbi.nlm.nih.gov/bioproject/PRJNA834074 | NCBI BioProject, PRJNA834074 |
| Marletaz F | 2023 | Comparative genomics of the sea urchin P. lividus highlights contrasting trends of genome and regulatory evolution in deuterostomes | https://www.ncbi.nlm.nih.gov/geo/query/acc.cgi?acc=GSE202034 | NCBI Gene Expression Omnibus, GSE202034 |

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
