## [Editor Report · eLife assessment]

This **valuable** study addresses the role of Rho-associated coiled-coil kinase (ROCK) and the cytoskeleton in the initiation and growth of the calcified endoskeleton of sea urchin embryos. Perturbation by two independent approaches (a morpholino and a selective inhibitor) provides **convincing** evidence that ROCK participates both in actomyosin regulation and in the gene regulatory network that controls skeletogenesis. Exciting areas of future work will be to elucidate the mechanisms by which ROCK influences gene expression and to further dissect the role of the cytoskeleton in mineralization.

---

## [Referee Report · Reviewer #1 (Public Review)]

Using a pharmacological and knock-down approach, the authors could demonstrate that ROCK activity is required for the normal development of the larval skeleton. The presence of ROCK in the pluteus stage depends on the activity of VEGF that is responsible for the formation of the tubular syncytial sheath of the calcifying primary mesenchyme cells in which the skeleton forms. The importance of ROCK in skeleton formation was confirmed in cell culture experiments, demonstrating that ROCK inhibition leads to decreased elongation and abnormal branching of spicules. µCT analyses underline this finding demonstrating that the inhibition of ROCK mainly affects elongation of spicules while growth in girth is little affected. F-actin labeling experiments could demonstrate that ROCK inhibition interferes with the organization of the actomyosin network in the early phase of skeleton formation, while f-actin organization in the tips of the elongating spicule is unaffected by the pharmacological inhibition of ROCK. Finally, ROCK inhibition strongly affects the expression of major regulatory and calcification-related genes in the calcifying cells. Based on these findings the authors propose a model for the regulatory interaction between the skeletogenic GRN, ROCK and the f-actin system relevant for skeletogenesis.

Comments on revised version:

In their manuscript Hijaze et al. adequately addressed the majority of my previous concerns in a satisfactory manner. In particular, they validated their morpholino knock-down experiments by explaining how they determined the optimal concentrations and provided an immunohistological evidence for the reduction in ROCK protein abundance. The authors also added new antibody stainings providing evidence that ROCK and F-actin do not interact directly but likely through other kinases that modulate f-actin, and that the localization of f-actin at the spicule tips remains unaffected by the knock-down. In addition, the authors revised their discussion to not overstate their observations, and by focusing on the potential mechanisms by which ROCK may affect biomineralization (i.e. mechano sensing and exocytosis of vesicles). Here I would like to add, that f-actin mediated exocytosis does not necessarily target mineral baring vesicles but may also promote the exocytosis of matrix proteins that are essential for the normal formation of the spicules and that are an integral component of other biominerals, as well. I strongly encourage the authors to continue on this exciting research, including the development of methods to analyze the molecular mechanisms that control vesicular trafficking in mineralizing systems.

---

## [Referee Report · Reviewer #2 (Public Review)]

This project is on the role of ROCK in skeletogenesis during sea urchin development. That skeleton is produced by a small number of cells in the embryo with signaling inputs from the ectoderm providing patterning cues. The skeleton is built from secretion of CaCO3 by the skeletogenic cells. The authors conclude that ROCK is involved in the regulation of skeletogenesis with a role both in regulating actomyosin in the process, and in the gene regulatory network (GRN) underlying the entire sequence of events.

The strength of the paper is that they show in detail how perturbations of ROCK results in abnormal actomyosin activity in the skeletogenic cells, and they show alterations both in expression of transcription factors of the GRN, and expression of genes involved in assembly of the skeletal matrix. Two different approaches lead to this conclusion: morpholino perturbations and the actions of a selective inhibitor of the kinase activity. Thus, they achieved their goal which was to test the hypothesis that ROCK is involved in the process of skeletogenesis. Those tests support the hypothesis with data that was quantitatively significant.

The discussion was transparent regarding where the analysis ended and where the next phase of work should begin. While actomyosin involvement was altered when ROCK was perturbed, it isn't known how direct or indirect the role of ROCK might be. Also, while the regulatory input to spicule initiation and growth is affected when ROCK is inhibited, it isn't clear exactly where ROCK is involved.

---

## [Author Response]

The following is the authors’ response to the previous reviews.

Thank you and the two reviewers for the thorough review of our manuscript. We thank you very much for the positive evaluation of our manuscript and your encouragement to continue in this fascinating topic. In this version we made minor changes in the text to address the comments and suggestion of the second reviewer and increase the clarity of the text.

**Reviewer #2 Recommendation to the authors**

We thank the reviewer for the sharp comments that help us improve the clarity of the paper. Below we list the changes we made to correct and revise the paper in accordance to the reviewer’s comments.

(1) Line 90. Isn't the genus Paracentrotus?

Yet it is, thank you. We corrected the typo.

(2) Figure 1 and supplementary figure 2. To this reviewer supplementary Figure 2 doesn't really help the story as written in the paragraph from line 96-110. You want to report expression of ROCK in skeletogenic cells. You do that quite well in Figure 1. Since Fig. S2 reports whole embryo expression of ROCK when only 5% of the cells in the embryo are the subject of interest here, and the Axitinib is selective, presumably for skeletogenic cells, the relative lack of effect in Fig. S2 is not surprising and again, doesn't really help the theme you wish to establish by focusing on the role of ROCK in skeletogenic cells over time. If anything, the data reported in Fig. S2 shows that perturbation of VEGF signaling has very little effect embryo-wide, while Fig. 1 shows that perturbation of VEGF signaling has a noticeable effect on ROCK expression in skeletogenic cells. If you choose to keep Fig. S2, I recommend that you indicate that embryo-wide vs skeletogenic cell difference more succinctly than given at present. It will also strengthen your paragraph in lines 110-127.

The importance of the western blot presented in Fig. S2 is to validate that the antibody recognizes a protein of the expected size. This strengthen the credibility of this commercial antibody to detect the sea urchin ROCK protein. We agree with the reviewer that the fact that the skeletogenic cells are less than 5% of the embryonic cells is important to explain why we didn’t see an affect of VEGFR inhibition in the western blot, and we changed the text to express it (lines 108-111): “Yet, this measurement was done on proteins extracted from whole embryos, of which the skeletogenic cells, where VEGFR is active, are less than 5% of the total cell mass (42). We therefore wanted to study the spatial expression of ROCK and specifically, its regulation in the skeletogenic cells.”

(3) Comparison of Fig. 2 and Fig. S3. To me the reader is confused when Fig. S3 is 33hpf as reported in the text (but not in the figure legend), and Fig. 2 shows 2 day old embryos - on the figure and figure legend but not in the text. So, the reader sees the text indicating 33hpf and looks around and the figure 2 says 2dpf. Does that mean 33hpf = 2dpf, the reader is thinking. To clarify, I suggest including the 2dpf in the text or simply drop the time in the text and report it in the two figures. Further, in the middle of the paragraph 130-143 you switch from reporting on Fig.S3 to Fig. 2, yet the reader doesn't know that. The reader is still looking at Fig. S3. The problem here is that at 33hpf the skeleton doesn't yet show the reduction or abnormalities that are shown later at 2dpf in Fig. 2. In clarifying this paragraph both the reduction in ROCK expression and the subsequent alterations in growth and patterning of the skeleton will be clear to the reader.

Thank you for raising this point. We added in the caption of Fig. S3 that the measurements were done in 33hpf. We also added in the text, that the observations of the skeletogenic phenotypes were done at 2dpf (48hpf). We made a break between the first paragraph discussing Fig. S3 and the paragraph discussing Fig. 2.

(4) The experiment with Y27632, an inhibitor of ROCK, is significantly improved in this revision. The concern earlier was the possibility that at the concentration used there might be off-target effects since other kinases are affected by higher concentrations of this selective inhibitor. The authors have modified this component of the paper and performed experiments at lower concentrations where other reports indicate the inhibitor is highly selective for ROCK, and they still demonstrate an inhibition of skeletal production. This, plus the added citations greatly increases confidence that this inhibition is selective for ROCK, thus enabling a stronger conclusion that ROCK has a role in skeletal growth and patterning.

Thank you for asking us to test this lower concentration which improved the credibility of our findings.

Line 239 - should be: indicating instead of indicting We corrected that.(5) Line 402-403."The first step in generating the sea urchin spicules is the construction of the spicule cavity, a membrane filled with calcium carbonate and coated with F-actin (Fig. 8A)". I suggest more precise language. The way this now reads (above) is that somehow the spicule cavity is a membrane and that membrane is filled with CaCO3. And further the membrane is coated with F-actin. Isn't the spicule cavity what is filled with CaCO3? And isn't that cavity surrounded by a membrane? And the F-actin must be in the cortex of the cell since there is very little cytoplasm associated with the pseudopodial extensions that surround the spicule.

We change this sentence to: “The first step in generating the sea urchin spicules is the construction of the spicule cavity where the mineral is engulfed in a membrane coated with F-actin” (lines 403-404). Our observations show that F-actin is enriched around the spicule cavity. It could be an extension of the cell cortex, but we did not prove it, so we prefer to simply describe what we saw.

Line 405-408. Thank you for putting in this unknown. It is important to point out that while you've shown that ROCK contributes to regulation of actomyosin, it is not clear whether this is direct or indirect. You have also shown that ROCK somehow contributes to regulation of the GRN that leads to skeletogenesis. Thus, your data are consistent in showing that ROCK perturbation cripples normal skeletogenesis both via morpholino and with a selective inhibitor. Your last part of the discussion then offers speculation as to what might be affected specifically. That discussion sets the stage for digging even deeper to identify specific targets of ROCK activity.

Thank you, we agree with you that there is an exciting road ahead of us!